# GENERATING ALL-ATOM PROTEIN STRUCTURE FROM SEQUENCE-ONLY TRAINING DATA

## ABSTRACT

Using generative models for protein design is gaining interest for their potential scientific impact. However, biological processes are mediated by many modalities, and simultaneous generating multiple biological modalities is a continued challenge. We propose **PLAID (Protein Latent Induced Diffusion)**, whereby multimodal biological generation is achieved by learning and sampling from the *latent space of a predictor* from a more abundant data modality (e.g. sequence) to a less abundant data modality (e.g. crystallized structure). Specifically, we examine the *all-atom* structure generation setting, which requires producing both the 3D structure and 1D sequence, to specify how to place sidechain atoms that are critcial to function. Crucially, since PLAID **only requires sequence inputs to obtain the latent representation during training**, it allows us to use sequence databases when training the generative model, thus augmenting the sampleable data distribution by $10^2\times$ to $10^4\times$ compared to experimental structure databases. Using sequence-only training further unlocks more annotations that can be used to conditioning model generation. As a demonstration, we use two conditioning variables: 2219 function keywords from Gene Ontology, and 3617 organisms across the tree of life. Despite not receiving structure inputs during training, model generations nonetheless exhibit strong performance on structure quality, diversity, novelty, and cross-modal consistency metrics. Analysis of function-conditioned samples show that generated structures preserve non-adjacent catalytic residues at active sites, and learn the hydrophobicity pattern of transmembrane proteins, while exhibiting overall sequence diversity. Model weights and code are publicly accessible at `[redacted]`.

## 1 INTRODUCTION

Generative protein models propose designs and can accelerate innovation in bioengineering. Many protein functions are mediated by their structure, including the identity, placement, and biophysical properties of both sidechain and backbone atoms, known as the *all-atom structure*. However, to know which sidechain atoms to place, one must first know the *sequence*; all-atom structure generation thus can be seen as a multimodal problem that requires simultaneous generation of sequence and structure.

While generative modeling for protein structures has seen rapid recent progress, important challenges remain: **(1)** Existing protein structure and sequence generation methods often treat sequence and structure as *separate modalities*; structure-generation methods often only provide backbone atoms. **(2)** Methods that do address all-atom design often require alternating between folding and inverse-folding steps using an extraneous

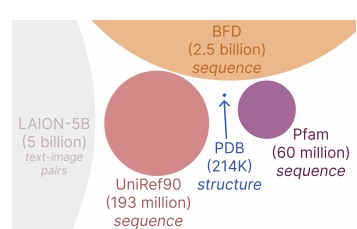

Figure 1: Drawn-to-scale size comparison of dataets. Compared to structural databases, sequence databases offer far more comprehensive coverage of the natural protein space than structure databases.

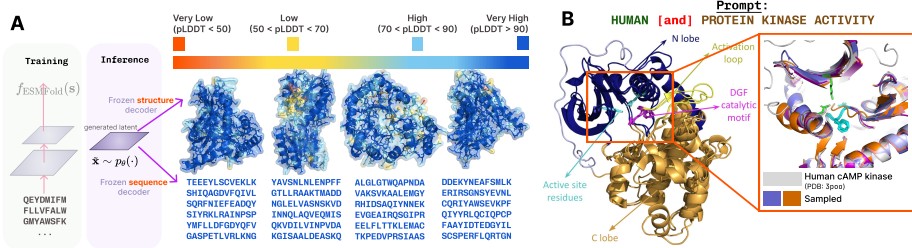

Figure 2: **(A)** Using the PLAID paradigm to sample from the latent space of ESMFold unconditionally generates high quality all-atom structure and sequence **despite using only sequence input** to train the generative model. **(B)** Since sequence-only databases has more annotations, we can compositionally **condition by function and expression organism**. Function-conditioned proteins can **preserve known catalytic residues**. An example is shown for generating human kinases; generations exhibit the DGF catalytic motif, and the global N-terminal and C-terminal lobes characteristic of human MAP kinases [1], while sharing only 48% global sequence identity to the generation. Generated samples are classified as being in active kinase conformation by the Kincore predictor [2].

prediction model. **(3)** Evaluations often emphasize *in silico* oracle-based designability, or structure-conditioning, with limited progress towards other forms of flexible controllability. **(4)** Methods that rely on experimentally-resolved structure databases have a strong bias towards crystallizable proteins. **(5)** Methods that ingest structure as inputs have more restrictions on architecture, and is harder to leveraging progress in hardware-aware Transformer models for more scalable large language models.

**Contributions** Towards resolving these challenges, we introduce **PLAID (Protein Latent Induced Diffusion)**. Our principal demonstration is that multimodal generation in biology can be achieved by learning the latent space of a predictor from a more abundant data modality (e.g. sequence) to a less abundant data modality (e.g. crystallized structure). In particular, we focus on ESMFold [3] and all-atom structure generation, and present a controllable diffusion model capable of **simultaneous sequence and all-atom structure generation, while requiring only sequence inputs during training**. Because training dataset can be defined by sequence databases rather than structural ones, one can obtain better coverage of the viable protein space traversed by evolution. It furthermore allows us to leverage structural information encoded in the *pretrained weights* rather than training data. Finally, it increases the availability of annotations for controllable generation. As a motivating demonstration, we examine compositional control across the axes of *function* and *organism*, though sequence databases also offer far more annotation types, such as natural language abstracts. The method is designed to be easily adaptable to expanding sequence datasets, capitalize on improved inference and training infrastructure for Transformer-based models [4, 5, 6], and increasingly multimodal protein folding models, such as nucleic acids and molecular ligand binding [7, 8].

## 2 RELATED WORKS

**Generative Modeling for Proteins** State-of-the-art diffusion models for designing protein structure have thus far focused on generating *novel backbone folds*, with conditioning controllability typically governed by secondary structure, or for generating scaffolding for a known motif [13, 14, 15, 16]. Evaluation and design of these models focus on fold stability and novelty, and often involve using oracle models [17, 3, 18, 19] for folding or inverse folding. However, to synthesize the protein, the sequence is required, and not all sampled structures might have a corresponding sequence. To address this, "designability" has been posited as a metric, which assesses the correspondence between the original structure and the sequence predicted for that structure.

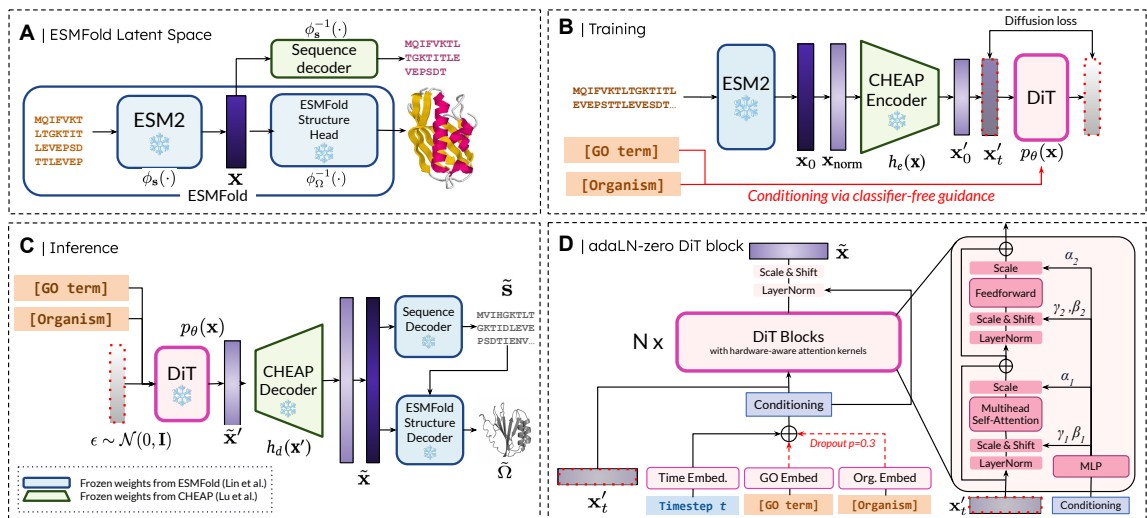

Figure 3: Overview of PLAID. **(A) ESMFold [3] latent space**. The latent space $p(\mathbf{x})$ can be considered a joint embedding of sequence and structure. **(B) Latent diffusion training.** Our goal is to learn and sample from $p_\theta(\mathbf{x})$, following the diffusion [9] formulation. To improve learning efficiency, we use the CHEAP [10] encoder $h_e(\cdot)$ to obtain compressed embedding $\mathbf{x}' = h_e(\mathbf{x})$, such that the actual diffusion objective becomes sampling from $p_\theta(h_e(\mathbf{x}))$. **(C) Inference.** To obtain both sequence and structure at inference time, we use the trained model to sample $\tilde{\mathbf{x}}' \sim p_\theta(\mathbf{x}')$, then uncompress using the CHEAP decoder to obtain $\tilde{\mathbf{x}} = h_d(\tilde{\mathbf{x}}')$. This $\tilde{\mathbf{x}}'$ embedding can then be decoded into amino acid identities using frozen sequence decoder (trained in CHEAP [10]). This is used as input along with $\tilde{\mathbf{x}}$ to the frozen structure structure decoder (trained in ESMFold [3]) to obtain the all-atom structure. **(D) DiT block architecture.** We use the Diffusion Transformer (DiT) [11] architecture, which uses adaLN-zero DiT blocks to incorporate conditioning information. Classifier-free guidance is used to incorporate the function (i.e. GO term) and organism class label embeddings [12].

However, there are few mechanisms to enforce designability during training. Methods also exist for designing sequence [20, 21, 22, 23], sometimes conditioned by the structure [24]. Structure can be constructed from these generations using a protein folding model, but models do not explicitly produce atomic positions.

**Multimodal Sequence-Structure and All-Atom Generation**    All-atom generation can thus be viewed as a multimodal generation problem, where the 1D protein sequence and 3D protein structure are jointly produced. Existing works [25, 26] often generate only one of structure or sequence at each diffusion step, and rely on an external predictor to produce the other modality. Multiflow [27] performs co-generation without an external tool, but does not produce side chain positions. Some works have focused on specific protein subclasses, such as antibody design [28, 29]. While these models achieve success within their specialized domains, antibodies represent a narrow subset of protein space and such models often struggle with out-of-distribution generalization to all protein families. Concurrently developed with this work, ESM3 [30] also uses generates in the shared sequence-structure space, and is conditioned on Interpro [31] (many of which are derived from GO terms [32]) for controllability. However, the ESM3 tokenizer is trained on structure datasets, rather than sequence databases, and cannot perform all-atom generation. It is easy to extend PLAID to the tokenized setting, since CHEAP [10] embeddings also includes a tokenized variant.

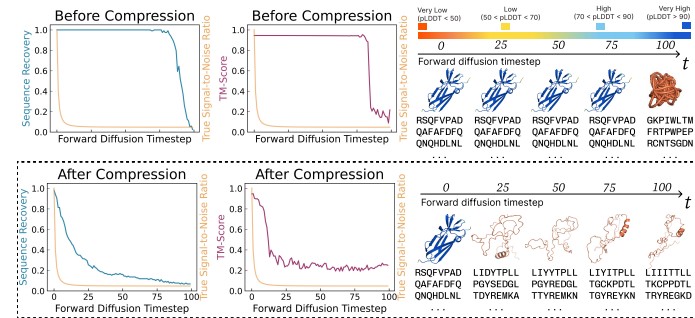

Figure 4: **Comparing latent space diffusion effects on sequence and structure.** We add noise via a cosine schedule [33] to both the raw, uncompressed latent space $\mathbf{x}$ and the compressed latent space, $\mathbf{x}' = h_e(\mathbf{x})$. The true signal-to-noise ratio (SNR) curve is overlaid in orange. Prior to the normalization and compression steps in CHEAP, noise added in the latent space does not affect sequence and structure until the final timesteps in forward diffusion, meaning that the denoising task would be trivial for most sampled timesteps. After adding the compression, corruptions in the sequence and structure space is closer to the true SNR.

## 3 PLAID: PROTEIN LATENT INDUCED DIFFUSION

**Notation** A protein is composed of component amino acids. A protein sequence $\mathbf{s} := \{r_i\}_{i=1}^{L}$ is often shown as a string of characters, with each character denoting the identity of an amino acid residue $r \in \mathcal{R}$, with $|\mathcal{R}| = 20$. Each unique residue $r$ can be mapped to a set of atoms as $\mathbf{r} := \{\mathbf{a}_i\}_i^M$, where $\mathbf{a} \in \mathbb{R}^3$ is the 3D coordinates of the atom, and the number of atoms $M$ in each residue $\mathbf{r}$ may be different depending on the identity. A protein structure $\Omega := \{\mathbf{r}_i\}_{j=1}^{L}$ consists of all atoms in the protein.[1]

**All-Atom Structure vs. Backbone-Only Structure** From above definitions, we see that the *all-atom structure* $\Omega$ requires knowledge of the amino acid identities at each position in order to specify the side chain atoms. To reduce complexity, protein structure designers sometimes work with the backbone atoms $\Omega_{\text{backbone}} \subset \Omega$ only, which only include the $N, C, C_\alpha$ atoms only, and are generally sufficient to define the protein fold.[2]

### 3.1 DEFINING $p(\text{SEQUENCE},\text{STRUCTURE})$

We begin with the motivation that sampling directly from $p(\mathbf{s}, \Omega)$ without implicitly factorizing it into $p(\Omega)p(\mathbf{s}|\Omega)$ (e.g., Protpardelle [25]) or $p(\mathbf{s})p(\Omega|\mathbf{s})$ (e.g., ProteinGenerator [26]) circumvents the difficulty in all-atom generation of not knowing which side chain atoms to place; one can choose a latent manifold where residues do not need to be explicitly specified during iterative generation. Avoiding reliance on external prediction tools is computationally cheaper, and avoids amplifying errors.

Our goal is to characterize a distribution $p(\mathbf{x})$ over $\mathcal{X}$ that encapsulates both sequence and structure information, such that there is a mapping $\mathbf{x} = \phi_{\mathbf{s},\Omega}(\mathbf{s}, \Omega)$. To do this, we follow the definition of joint embedding

---

[1]In practice, to make use of array broadcasting, a standard $M$ is selected for all residues, with an associated one-hot mask to specify which atoms are present for a given residue, and we treat each structure as a matrix $\Omega \in \mathbb{R}^{L \times M \times 3}$. Following prior work [34, 3], we use the atom14 representation where $M = 14$.

[2]The three torsion angles in backbone-only structures induces $3^L$ degrees of freedom; depending on the residue identity, there may be 0 to 4 additional rotamer angles associated with the sidechains. Therefore, even when the sequence is known, there may up to $4^L$ additional degrees of freedom necessary for all-atom structure prediction.

of sequence and structure in Lu et al. [10]: if we decompose $\mathbf{x} = \phi_{\mathbf{s},\Omega}(\mathbf{s}, \Omega) = \phi_{\mathbf{s}}(\mathbf{s}) \circ \phi_{\Omega}(\Omega)$, we can look for a space where some deterministic mapping will map sequence $\mathbf{s}$ and its corresponding structure $\Omega$ to the same latent embedding $\mathbf{x} \in \mathcal{X}$. One way to do so is by defining $\mathbf{x}$ as the latent space of a protein folding model $p(\Omega|\mathbf{s})$. The trunk of the model provides $\mathbf{x} = \phi_{\text{ESM}}(\mathbf{s})$, and the structure head provides $\Omega = \phi_{\text{Structure Module}}(\mathbf{x})$. If we consider there to be an implicit inverse function of the Structure Module such that $\mathbf{x} = \phi_{\text{Structure Module}}^{-1}(\Omega)$, then this provides the mappings for $\mathbf{x} = \phi_{\mathbf{s},\Omega}(\mathbf{s}, \Omega) = \phi_{\mathbf{s}}(\mathbf{s}) \circ \phi_{\Omega}(\Omega)$ that we are looking for.

**Overview of ESMFold** Briefly, ESMFold [3] has two main components: a protein language model component $\mathbf{x} = \phi_{\text{ESM2}}(\mathbf{s})$ that captures evolutionary priors via the masked language modeling loss (MLM), and a structure module component $\Omega = \phi_{\text{Structure Module}}(\mathbf{x})$ that decodes these latent embeddings into a 3D structure. For the rest of this work, "latent space of ESMFold" refers to the $\mathbf{x} \in \mathbb{R}^{L \times 1024}$ representation at the layer just prior to the Structure Module, where $L$ is the length of a given protein. We choose this layer due to the observations in Lu et al. [10] (also see Section 3.2) that the pairwise input at inference time to the Structure Module is initialized to zeros, such that this sequence embedding contains all information for structure prediction (Figure 3A and Appendix B).

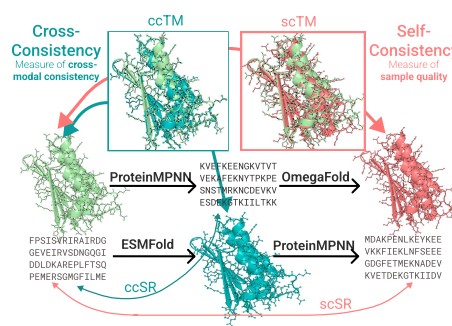

Figure 5: Schematic of consistency metrics used for evaluation (Section E).

## 3.2 SAMPLING ALL-ATOM STRUCTURE

**Latent Generation** Our goal is to learn $p_\theta(\mathbf{x}) \approx p(\mathbf{x})$, where $\theta$ are parameters of the model learned through diffusion training (Figure 3C). Then, after training, we can sample $\tilde{\mathbf{x}} \sim p_\theta(\mathbf{x})$ (Figure 3C). To do so, we use diffusion models [9, 35], with some modifications (described in ablation Table 1). To obtain structure from the sampled latent embedding, we can simply use frozen ESMFold structure module weights to obtain $\tilde{\Omega} = \phi_{\text{Structure Module}}(\tilde{\mathbf{x}})$ (Figure 3C). Since the output of $\phi_{\text{Structure Module}}$ is all-atom, the sampled $\tilde{\Omega}$ is also all-atom.

**Sequence Decoder** To obtain the sequence, we need an "inverse mapping of ESM2" to get $\tilde{\mathbf{s}} = \phi_{\text{ESM}}^{-1}(\tilde{\mathbf{x}})$. This inverse mapping is straightforward to train, since $\mathbf{x}$ is a linearly projected version of the ESM2 embedding, which was trained via the MLM loss. This sequence decoder $\phi_{\text{ESM}}^{-1}$ is also trained and provided in Lu et al. [10], with validation accuracy on a heldout partition of UniRef [36] reaching 99.7% [10]. Note that $\tilde{\mathbf{s}}$ must be decoded first, which determines the side-chain atoms to be placed in $\tilde{\Omega}$.

**Latent Space Compression** In initial experiments, we found that directly learning $p(\mathbf{s})$ performed poorly (results shown in Appendix Figure 10). We suspected that this might be due to the dimensions of $\mathbf{x} \in \mathbb{R}^{L \times 1024}$. For proteins with length $L = 512$ (the length cut-off used in this work), this maps to a high-resolution synthesis problem in image diffusion literature. We therefore mirror works in this literature, and perform diffusion in the latent space of an autoencoder $\mathbf{s}' = h_e(\mathbf{x})$ such that the array dimensions of $\mathbf{x}'$ is much smaller [37]. We use the CHEAP autoencoder [10], such that diffusion training becomes $p_\theta(\mathbf{x}') = h_e(\mathbf{x})$. Noise is added and denoised from $p(\mathbf{x}')$. At inference time, we first sample the compressed latent $\tilde{x}' \sim p_\theta(\mathbf{x}')$, then "uncompress" it to $\tilde{\mathbf{x}} = h_d(\mathbf{x}')$, followed by using frozen decoders to obtain $\tilde{\mathbf{s}} = \phi_{\text{ESM}}^{-1}(\tilde{\mathbf{x}})$ and $\tilde{\Omega} = \phi_{\text{Structure Module}}(\tilde{\mathbf{x}})$. Figure 4 illustrates how noised added in the latent space manifests in the sequence and structure spaces, and Appendix Figure 10 shows results without compression. More information on CHEAP can be found in Lu et al. [10] and Appendix B.

### 3.3 DATA AND TRAINING

**Choice of Sequence Database**    The general paradigm in PLAID can be used on any sequence database. As of 2024, sequence-only database sizes can range from UniRef90 [36] (193 million sequences) to metagenomic datasets such as BFD [38] (2.5 billion sequences) and OMG [39] (3.3 billion sequences). We use Pfam because it provides more annotations for *in silico* evaluation, and because protein domains are the main units of structure-mediated functions. More information can be found in Appendix C.

**Compositional Conditioning by Function and Organism**    Gene Ontology (GO) is a structured hierarchical vocabulary for annotating gene functions, biological processes, and cellular components across species [40, 41]. We examine all Pfam domains for which there exists a Gene Ontology mapping; there are 2219 GO terms compatible with our model (an abbreviated list is listed in Appendix **??**). We also examine all unique organisms in our dataset, and find 3617 organisms. Models are trained with classifier-free guidance [12]. The conditioning architecture is described in Figure 3D. More details can be found in Appendix A.

**Architecture**    We use a Diffusion Transformer [11] (DiT) for the denoising task. This enables more flexible options for finetuning on mixed input modalities, as protein structure prediction models begin expanding to complexes with nucleic acids and small molecular ligands. It also makes better use of Transformer training infrastructure [42, 4, 43, 5, 44]. In early

Table 1: Ablation results for metrics defined in Section 4.

|   | Configuration | ccTM | scTM | Ppl. | Seq. Div. % | Struct. Div.% |
|---|---|---|---|---|---|---|
| A | cosine noise sched.& pred. noise | 0.54 | 0.55 | 16.97 | **0.98** | 0.86 |
| B | A + v-diffusion | 0.52 | 0.53 | 17.37 | **0.98** | **0.89** |
| C | A + MinSNR | 0.59 | 0.59 | 16.76 | 0.97 | 0.86 |
| D | A +B + C + sigmoid noise sched. | 0.56 | 0.58 | 16.88 | 0.92 | 0.86 |
| E | D + self-conditioning | **0.70** | **0.65** | **15.38** | 0.93 | 0.76 |
| F | E + no cond drop | 0.57 | 0.57 | 17.28 | 0.97 | 0.85 |

experiments, we found that proportioning available memory to a larger DiT model was more helpful than using triangular self-attention [17]. We train our models using the xFormers [42] implementation of [45], which provided a 55.8% speedup with a 15.6% reduction in GPU memory usage in our inference-time benchmarking experiments compared to a vanilla implementation using PyTorch primitives (Appendix G). We train two versions of the model with 100 million and 2 billion parameters respectively, both for 800K steps. More details are in Appendix A.

**Diffusion Training and Inference-Time Sampling**    We use the discrete-time diffusion definition proposed in Ho et al. [9], using 1000 timesteps. Additional strategies are used to stabilize training and improve performance: min-SNR reweighting [46], v-diffusion [47, 48], self-conditioning [49, 50], and a Sigmoid noise schedule [51], and EMA (exponential moving average) decay. Ablation results are shown in Table 1. For sampling, unless otherwise noted, all results use the DDIM sampler [33, 35] with 500 timesteps. We use $c = 3$ as the conditioning strength for conditional generation; however, we find (Appendix Figure 14C) that sample quality is not strongly affected by this hyperparameter. We also find that that DPM-Solvers [52] can attain comparable results with $10\times$ fewer steps in cases where speed is of concern (Appendix Figure **??**), but here prioritize sample quality. More details are in Appendix D.

## 4 EVALUATION

Following previous works and to address the unique challenges of all-atom generation, we examine the following metrics (more details in Appendix E). A schematic of consistency metrics is shown in Figure 3.1

1. **Multimodal Cross-Consistency:** When the generated sequence is folded, does it match the generated structure? *[Cross-consistency TM-Score (**ccTM**), cross-consistency RMSD (**ccRMSD**).]* When the generated structure is inverse-folded into a sequence, does it match the generated sequence? *[Cross-consistency sequence recovery (**ccSR**).]* What percentage of generated samples are designable? *[**ccRMSD < 2Å**.]*

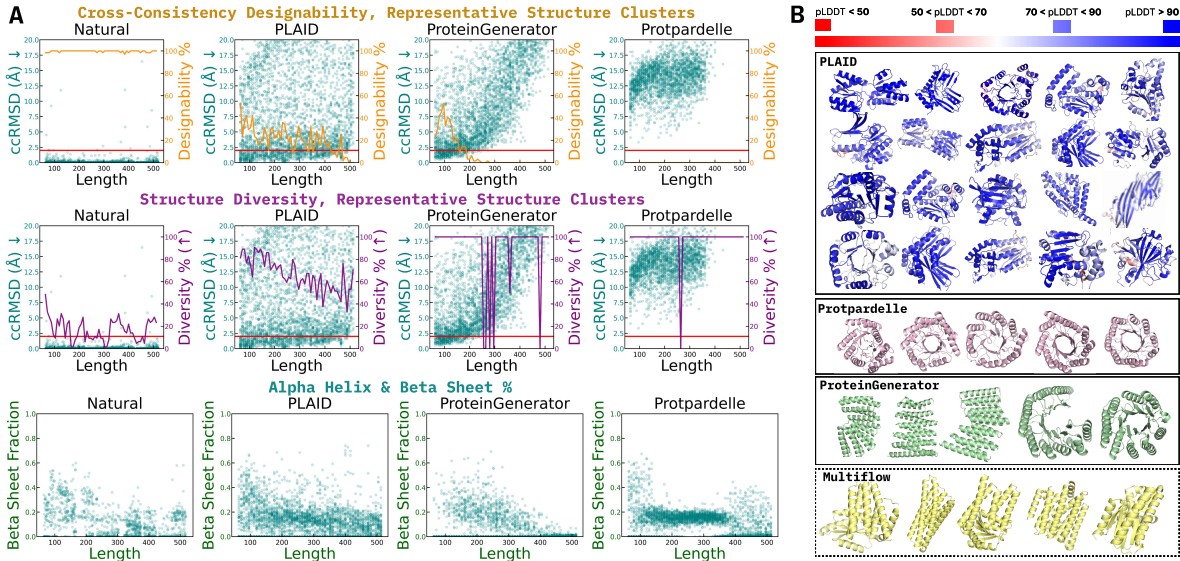

Figure 6: **By-length analysis of quality, sample diversity, and secondary structures**. Additional Figures can be found in Appendix Figure 11. **(A)** At each protein length, we plot: **1.** The fraction of designable samples that have ccRMSD < 2Å; samples with ccRMSD > 20Å are not plotted. The red line denotes the ccRMSD= 2Å threshold. **2.** The ratio of unique structure clusters to samples, as a measure of structural diversity; **3.** The beta sheet and alpha helix percentage of generations. At higher sequence lengths, PLAID can produce higher quality samples, whereas baseline methods often struggle, and/or exhibit mode collapse. **(B)** Unconditional generation results on proteins with length 256 using PLAID. Protpardelle [25] and ProteinGenerator [26] suffer mode collapse at this length towards TIM barrels and alpha helix bundles.

2. **Uni-modal Sample Quality**: What is the quality of generated sequence and structure, when examined independently?

   (a) *Structure.* If we inverse fold a generated structure into a sequence, and fold the result with OmegaFold [53], is it consistent with the original? *[Self-consistency TM-Score (scTM), self-consistency RMSD (scRMSD).]*

   (b) *Sequence.* If we fold a generated sequence, and inverse-fold the result, is it consistency with the original? *[Self-consistency sequence recovery (scSR).]* Do generated sequences have low perplexity on next-token prediction models trained on natural proteins? *[Perplexity (Ppl.) under RITA XL [23].]*

3. **Naturalness**: Do samples exhibit sensible biophysical parameters for real-world characterization? In other words, how similar are the distributions of biophysical properties between generated proteins and real proteins? *[Distributional conformity [20] scores.]*

4. **Diversity**: After clustering [54, 55], how many distinct and designable clusters can we observe? *[# Des. seq. clusts., # Des. struct. clusts..]*

5. **Novelty**: How similar generated structures to its closest structural match to real proteins? *[Foldseek TMScore.]* What about sequences? *[MMseqs seq id. %.]*

# 5 EXPERIMENTS

Table 2: Comparison of model performance across **consistency and quality metrics**. Bold values show best performance among all-atom generation models. pLDDT refers to the confidence score directly returned by the structure trunk of the generative model; for models which do not return a PLDDT metric, N/A is used. Heavy asterisk (*) indicates models which generates backbone structure and residue identities without sidechain positions.

| | Cross-Modal Consistency | | | | Structure Quality | | | Sequence Quality | |
|---|---|---|---|---|---|---|---|---|---|
| | ccTM (↑) | ccRMSD (↓) | ccSR (↑) | % ccRMSD < 2Å (↑) | scTM (↑) | pLDDT (↑) | Beta sheet % (↑) | scSR (↑) | Ppl. (↓) |
| ProteinGenerator | 0.58 | 11.86 | **0.28** | 0.08 | **0.72** | **69.00** | 0.04 | 0.40 | **8.60** |
| Protpardelle | 0.44 | 24.28 | 0.22 | 0.00 | 0.57 | N/A | 0.11 | **0.44** | 8.86 |
| PLAID | **0.69** | **9.47** | 0.26 | **0.32** | 0.64 | 59.46 | **0.13** | 0.27 | 14.61 |
| **Multiflow*** | 0.92* | 2.45* | 0.52* | 0.78* | 0.91* | N/A | 0.10* | 0.61* | 8.1* |
| *Natural* | *1.00* | *0.07* | *0.39* | *1.00* | *0.84* | *84.51* | *0.13* | *0.39* | *7.40* |

Table 3: **Diversity, novelty, and distributional conformity [20]** metrics across models. Asterisk (*) indicates methods which do not generate sidechain positions and bold indicates best performance across all-atom generation methods.

| | Diversity | | | Novelty | | Distributional Conformity (Wasserstein Distance) | | | | | |
|---|---|---|---|---|---|---|---|---|---|---|---|
| | # Des. (↑) | # Des. Seq. Clusts. (↑) | # Des. Struct. Clusts. (↑) | MMseqs Seq Id % (↓) | Foldseek TMScore (↓) | Avg. MW. (↓) | Aroma-ticity (↓) | Instab-ility Index (↓) | Iso-electric Point (↓) | GRAVY (↓) | Charge pH=7 (↓) |
| PG | 309 | 309 | 309 | 0.57 | **0.57** | 9.54 | 0.07 | 14.55 | 1.42 | 0.31 | 6.12 |
| Protpardelle | 0 | 0 | 0 | **0.56** | 0.72 | 10.4 | 0.07 | 8.61 | 1.99 | 0.37 | 8.58 |
| PLAID | **1171** | **809** | **522** | 0.60 | 0.67 | **0.62** | **0.01** | **1.98** | **0.49** | **0.28** | **2.71** |
| Multiflow* | 2812* | 2452* | 460* | 0.45* | 0.68* | 5.43* | 0.07* | 4.11* | 1.59* | 0.3* | 7.55* |
| *Natural* | *3570* | *1362* | *600* | *0.81* | *0.87* | *0* | *0* | *0* | *0* | *0* | *0* |

## 5.1 UNCONDITIONAL GENERATION

Following prior work demonstrating the effect of protein length on performance [13, 25, 27], we sample 64 proteins for each protein length between $\{64, 72, 80, ..., 496, 504, 512\}$, for a total of 3648 samples. Results in Figure 6 and Tables 2 and 3 show that while the degradation in performance for PLAID is far less pronounced than other all-atom methods. At longer lengths, PLAID can better balance quality and diversity. This may be due to the fact that the expanded dataset means that there are more samples available for lengths that are less commonly seen in the dataset. Despite not seeing structures when training the diffusion model, PLAID is able to achieve high cross-modal consistency between generated sequences and structures. Table 3 shows that the distribution of biophysical features for PLAID generations are closer to that of natural proteins, potentially due to the removed biases towards structure in its training data.

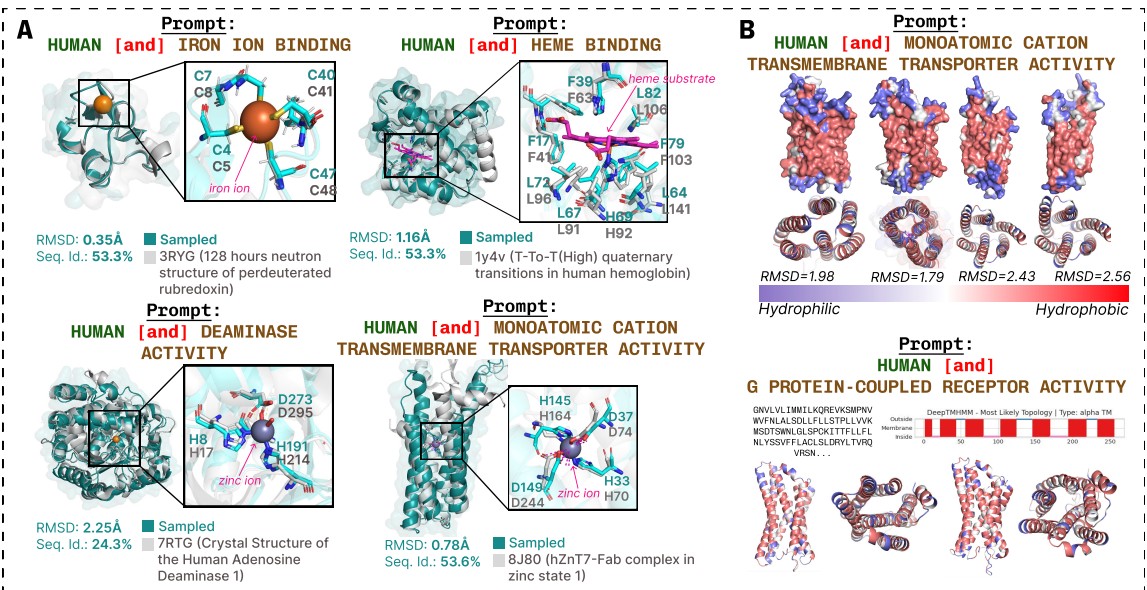

Figure 7: **Function conditioned generation for human proteins.** More examples are in Appendix 12. **(A) PLAID generations preserve catalytic motifs at non-adjacent residues, despite maintaining low sequence identity.** For each generation, we examine the closest Foldseek neighbor in the PDB [56] that was crystallized in complex with a ligand, and structurally align the generated sample to examine the active site. **(B) Generated membrane proteins recapitulate known hydrophobicity patterns**. *(Top)* Generated samples consistently match known hydrophobicity patterns of membrane proteins. Hydrophobic residues are found at transmembrane portions that span the lipid bilayer, while hydrophilic residues facing the aqueous environment. *(Right)* GPCR samples exhibit the expected 7-helix structure. DeepTMMHMM [57] predictions on sequences classifies generations as alpha transmembrane proteins, matching the known topology of GPCRs.

## 5.2 CONDITIONAL GENERATION

Computational evaluation of function- and organism-conditioned generative models presents a conundrum: lower similarity is a favorable heuristic in machine learning, since it is indicates that the generative model did not merely memorize the training data. From a bioinformatics perspective, however, conservation is key to function; taxonomic membership can be difficult to validate, given the high degree of similarity between homologs. In our case study experiments, we look for **high structural similarity to evaluate for function conditioning**, and **low sequence similarity to penalize exact memorization**. Case studies shown in Figure 7 show that function-conditioned proteins possess known biological characteristics, such as conserved active site motifs, and membrane hydrophobicity patterns. Global sequence diversity is low despite high levels of conservation at catalytic sites, suggesting the the model has learned key biochemical features associated with the function prompt without direct memorization. Appendix Figure 14 considers how conditioning scale might affect sample quality, and possible GO term characteristics that might be influencing the difference in Sinkhorn Distance between function-conditioned generations and random proteins.

We further by examining the Sinkhorn distance between generated latent embeddings and real proteins from **heldout** validation set unseen during training (Figure 8). This assess conditional generations indepedent of the sequence and structural decoders. For comparison, the Sinkhorn distance between random real proteins

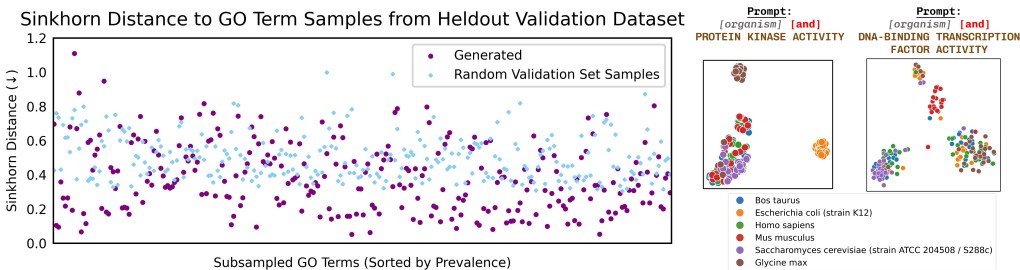

Figure 8: **(Left)** For each unique GO term in the validation set, we examine the Sinkhorn distance between generated samples and real proteins in the heldout subset in this class. For reference, we also calculate the Sinkhorn distance between random real proteins to the heldout subset. **(Right)** T-SNE reduction of generated embeddings, colored by the organism used for conditioning.

from the validation set and the function-conditioned generations are also evaluated. Conditional generations generally have lower Sinkhorn distances than random samples, suggesting that the desired latent information has been captured in the embedding. Figure 8 shows tSNE plots of generated embeddings colored by organism. Organisms that are further away phylogenetically, such as Glycine max (i.e. soybean) and E. coli, form more distinct clusters than those closer evolutionarily, such as human and mouse. This suggests that function and organism conditioned samples have been imbued with desired characteristics at a high level.

## 6 DISCUSSION

We proposed PLAID, a paradigm for multi-modal, controllable generation of proteins by diffusing in the latent space of a prediction model that maps single sequences to the desired modality. Our method is designed to adhere to progress in **data availability, model scalability, and sequence-to-structure prediction capabilities**. To this end, we chose an implementation that can make use of fast attention kernels [42] for Transformer-based architectures, and chose GO terms as a proxy for the vast quantities of language annotation that are paired with sequence databases.

It is straightforward to expand PLAID to many downstream capabilities. First, though we do not examine motif scaffolding or binder design explicitly in this current work, this is easy to build by holding some input residues constant. Second, though we examine ESMFold [3] in this work, the method can be applied to any prediction model. There is rapid progress [8, 7, 58, 59, 60] in predicting complexes from structure, and diffusing in the latent space of such models enables us to use the frozen decoder to obtain more modalities than just all-atom structure.

A limitation of PLAID is that performance can be bottlenecked by the prediction model from which the frozen decoders are derived. Here, we rely on the optimism that such models will continue to improve. With explicit finetuning for latent generation (e.g. training CHEAP and the structure decoder end-to-end), model performance can likely be improved. Furthermore, since the current structure decoder is deterministic, it does not sample alternative conformations. A solution is to use a decoder that returns a distribution over structural conformations instead; such a model might naturally be developed with progress in the field, or be explicitly finetuned for. Additionally, the GO term one-hot encoding used here does not take into account the hierarchical nature of the Gene Ontology vocabulary, nor that a protein might have several relevant GO terms. Finally, the classifier-free guidance scale can be separated for the organism and function conditions, since the two may require different guidance strengths in real-world use cases. These limitation will be examined in future work.

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

APPENDIX

# A ADDITIONAL TRAINING DETAILS

We train two variants of the model: a 2B version and a 100M version, both with the memory efficient attention implementation in xFormers, with float32 precision. A learning rate of 1e-4 was used, with a cosine annealing applied over 800,000 steps. The xFormers memory efficient attention kernel requires input lengths to be a multiple of 4. Since we also apply an upsampling factor of 2, the actual inference length must be a multiple of 4. During training, the maximum sequence length we use is 512, based on the distribution of sequences in Pfam and a shorten factor of 2 based on results in Lu et al. [10].

Following Ho and Salimans [12], with $p_{\text{uncond}} = 0.3$, the class label is replaced with the $\varnothing$ unconditional token. This is sampled separately for both function and organism. Note that not all data samples will have an associated GO term; we use the $\varnothing$ token for those cases as well. At inference time, to perform generate unconditionally (for either or both of function and/or organism), we use the $\varnothing$ token for conditioning.

# B CHEAP COMPRESSION DETAILS

Briefly, the CHEAP encoder and decoder uses an Hourglass Transformer [61] architecture that downsamples lengthwise, as well as downprojects the channel dimension, to create a bottleneck layer, the output of which is our compressed embedding. The entire model is trained with the reconstruction loss $MSE(\mathbf{x}, \hat{\mathbf{x}})$. Results in Lu et al. [10] show that structural and sequence information in ESMFold latent spaces are in fact highly compressible, and despite using very small bottleneck dimensions, reconstruction performance can be nonetheless maintained when evaluated in sequence or structure space.

Based on reconstruction results in Lu et al. [10], we choose $\mathbf{x}' \in \mathbb{R}^{\frac{L}{2} \times 32}$ with $L = 512$, which balances reconstruction quality at a resolution comparable to the size of latent spaces in image diffusion models [37]. Dividing the length in half allows us to better leverage the scalability and performance of Transformers, while managing its $\mathcal{O}(L)$ memory needs.

The CHEAP module involves a channel normalization step prior to the forward pass through the autoencoder. We find that the distribution of embedding values is fairly "smooth" here (Figure 9). Though the original Rombach et al. [37] paper was trained with a KL constraint to a Gaussian distribution, we use the embedding output as is. CHEAP embeddings were also trained with a $tanh$ layer at the output of the bottleneck; this allows us to clip our samples between $[-1, 1]$ at each diffusion iteration, as was done in original image diffusion works [9, 12, 33, 62]. We found in early experiments being able to clip the output values were very helpful for improving performance. Without using the CHEAP compression prior to diffusion, sample quality was poor, even on short ($L = 128$) generations, shown in Figure 10.

# C DATA

We use the September 2023 Pfam release, consisting of dataset, consisting of 57,595,205 sequences and 20,795 families. PLAID is fully compatible with larger sequence databases such as UniRef or BFD (roughly 2 billion sequences), which would offer even better coverage. We elect to use Pfam because sequence domains have more structure and functional labels, which is easier for *in silico* evaluation of generated samples. We also hold out about 15% of the data for validation.

Approximately 46.7% of the dataset (N=24,637,236) is annotated with a GO term. Using the publicly available mapping as of July 1, 2024, we take a count of all GO occurrences; for each Pfam entry with

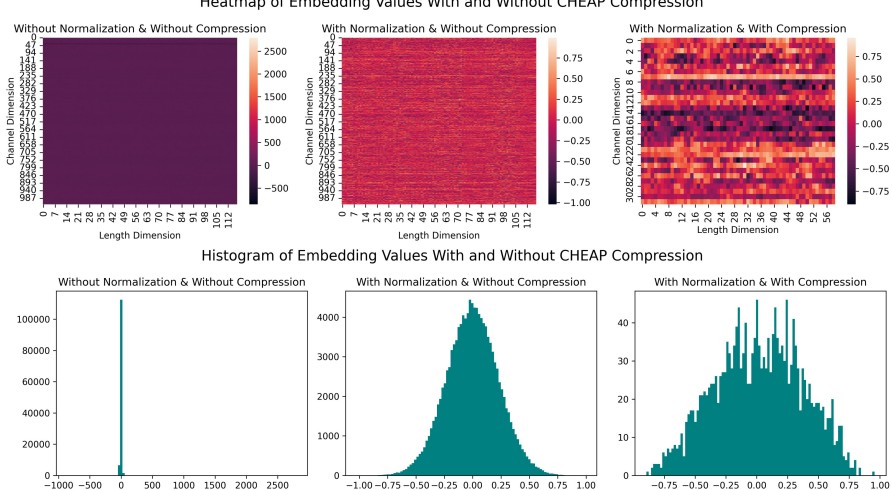

Figure 9: Visualizing the original ESMFold latent space before normalization, after per-channel normalization, and after compression. The value distribution of $p(\mathbf{x}')$ is fairly smooth and "Gaussian-like", making it amenable to diffusion.

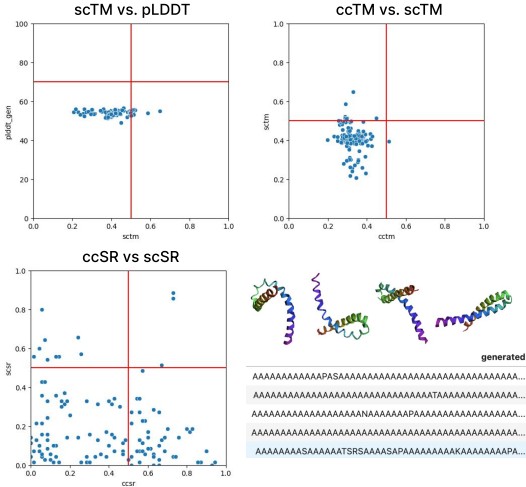

Figure 10: Results when running PLAID on the ESMFold latent space naively without CHEAP compression, for proteins of length 128. There is a tendency to generate repeated sequences, and quality is low compared to baselines.

multiple GO entries, we pick the one with the fewest GO occurences to encourage more descriptive and distinct GO labels.

The `Pfam-A.fasta` file available from the Pfam FTP server includes the UniRef code of the source organism from which the Pfam domain is from. The UniRef code further more includes a 5 letter "mneumonic" to denote the organism. We examine all unique organisms in our dataset, and find 3617 organisms.

# D   SAMPLING

Inference-time sampling hyperparameters provides the user with additional control over quality and sampling speed trade-off. PLAID supports the DDPM sampler [9] and the DDIM sampler [33], as well as the improved speed samplers from DPM++ [52]. We find that using the DDIM sampler with 500 timesteps using either the sigmoid or cosine schedulers works best during inference, and reasonable samples can be obtained using the DPM++2M-SDE sampler with only 20 steps. Experiments shown here uses DDIM sampler with the sigmoid noise schedule at 500 timesteps.

Note that the performance bottleneck is found mostly during the latent sampling and structure decoding (which depends on the number of recycling iterations [17, 3] used); however, these two processes can be easily decoupled and parallelized, which cannot be done in existing protein diffusion methods. Furthermore, it allows us to prefilter which latents to decode using heuristic methods, and decode only those latents to structure, which would boost performance for nearly the same computational cost. We do not empirically explore this in this paper to provide a fair comparison, and because the filtering criteria would vary greatly by downstream use.

# E   EVALUATION DETAILS

For all benchmarks and models, we use default settings provided in their open-source code. For ProteinMPNN [18], we use the `v_48_002` model with a sampling temperature of 0.1 and generate 8 sequences for protein, from which the best performing sequence use chosen. To calculate self-consistency, we fold sequences using OmegaFold [53] rather than ESMFold.

Though our models generate all-atom structure, we examine $C_\alpha$ RMSD rather than all-atom RMSD, to avoid mis-attributing sequence generation under-performance to structure generation failures. Also, since there are usually differences in the sequence that is generated, different number of atoms make it difficult to assess all-atom RMSD.

For the hold-out natural reference dataset, we use sequences from Pfam and keep length distributions similar to that of the sampled proteins. Specifically, for each sequence bin between $\{64, 72, ..., 504, 512\}$, we take 64 natural sequences of that length. For the experiment in Figure 14D, we use the Sinkhorn Distance rather than the Frechet Distance used commonly in images and video. Since not all functions have a large number of samples, we elected to use a metric that works better in low sample settings.

Structure novelty is obtained by searching samples to PDB100 using Foldseek [55] `easy-search`. We examine the TM-score to the closest neighbor. For Foldseek and MMseqs experiments, all clustering experiments are performed by length. We use default settings for both tools. Though we report the average TMScore to top neighbor for Foldseek, we run `easy-search` in 3Di mode. For sequences, we use MMseqs2 [54] to see if sequences have a homolog in UniRef50, using default sensitivity settings. For samples with homologs, we further calculate the average sequence identity to the closet neighbor to assess novelty (Seq ID %).

## F    SAMPLING SPEEDS

We examine the amount of time necessary for generating a simple sample. We first explore the time necessary to generate 100 sequences with $L = 600$. Multiflow and ProteinGenerator does not support batched generation in its default implementation, so in this experiment, we simply generate one sample at a time for a total of 100 samples. We report the amount of time per sample. For comparison, we also run an experiment where we only generate a single sample, such that none of the methods can make use of any improvements from batching.

Table 4: Time required to sample **proteins with 600 residues**. We assess time required both for sampling $N = 100$ samples in batches whenever possible, and when generating a single sequence. Experiments are run on Nvidia A100. Methods marked by (*) do not support batching in the default implementation.

|  | seconds/sample, batched | | seconds/sample, unbatched | |
|---|---|---|---|---|
|  | Sample Latent | Decode | Sample Latent | Decode |
| Protpardelle | 11.21 | - | 17.16 | - |
| Multiflow* | 231.32 | - | 277.11 | - |
| ProteinGenerator* | 343.32 | - | 342.28 | - |
| **PLAID (100M)** | 1.64 | 15.12 | 27.63 | 1.07 |
| **PLAID (2B)** | 19.34 | 15.07 | 49.03 | 0.9 |

## G    ATTENTION SPEED

Forward pass benchmark of vanilla multihead attention compared to the optimized xFormers implementation of memory-efficient attention [45] and FlashAttention-2 [4]. Though FlashAttention2 performed best in our benchmarks, a fused kernel implementation with key padding was not yet available at the time of writing. Since our data contained different lengths (as compared to most image diffusion use cases, or language use-cases that can make use of the implemented causal masking), we instead use the xFormers implementation. We expect that sampling speed results would improve once this feature is becomes available in the FlashAttention package.

| Method | Mean Time (s) | Mean Memory (GB) |
|---|---|---|
| Standard Multihead Attention | 0.0946 ± 9.23e-4 | 76.0 ± 0.409 |
| xFormers Memory Efficient Attention | 0.0519 ± 4.33e-05 | 64.0 ± 0.409 |
| Flash Attention | 0.0377 ± 1.91e-3 | 49.2 ± 0.783 |

## H    ADDITIONAL RESULTS

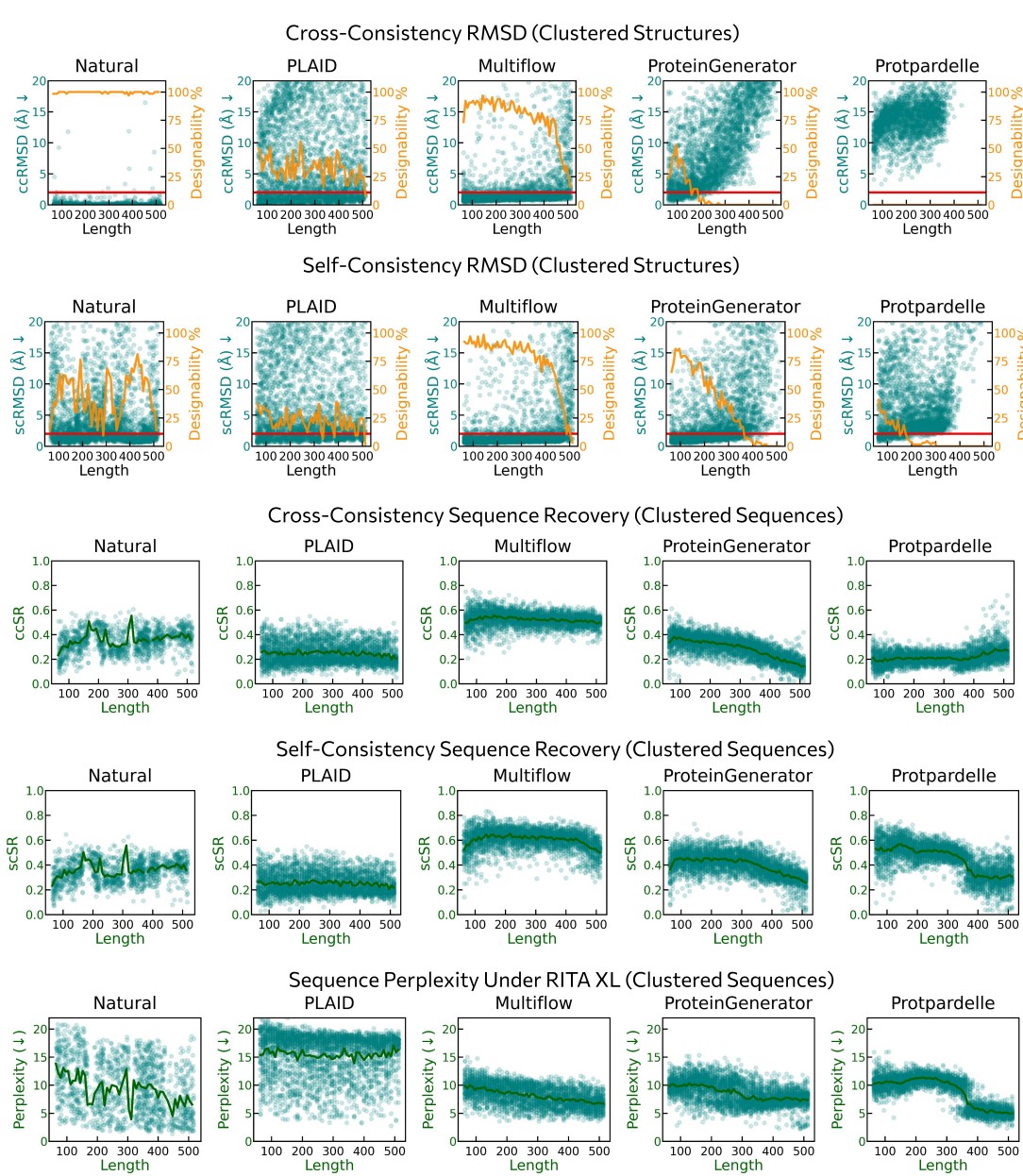

Figure 11: More comparison results between PLAID and baselines.

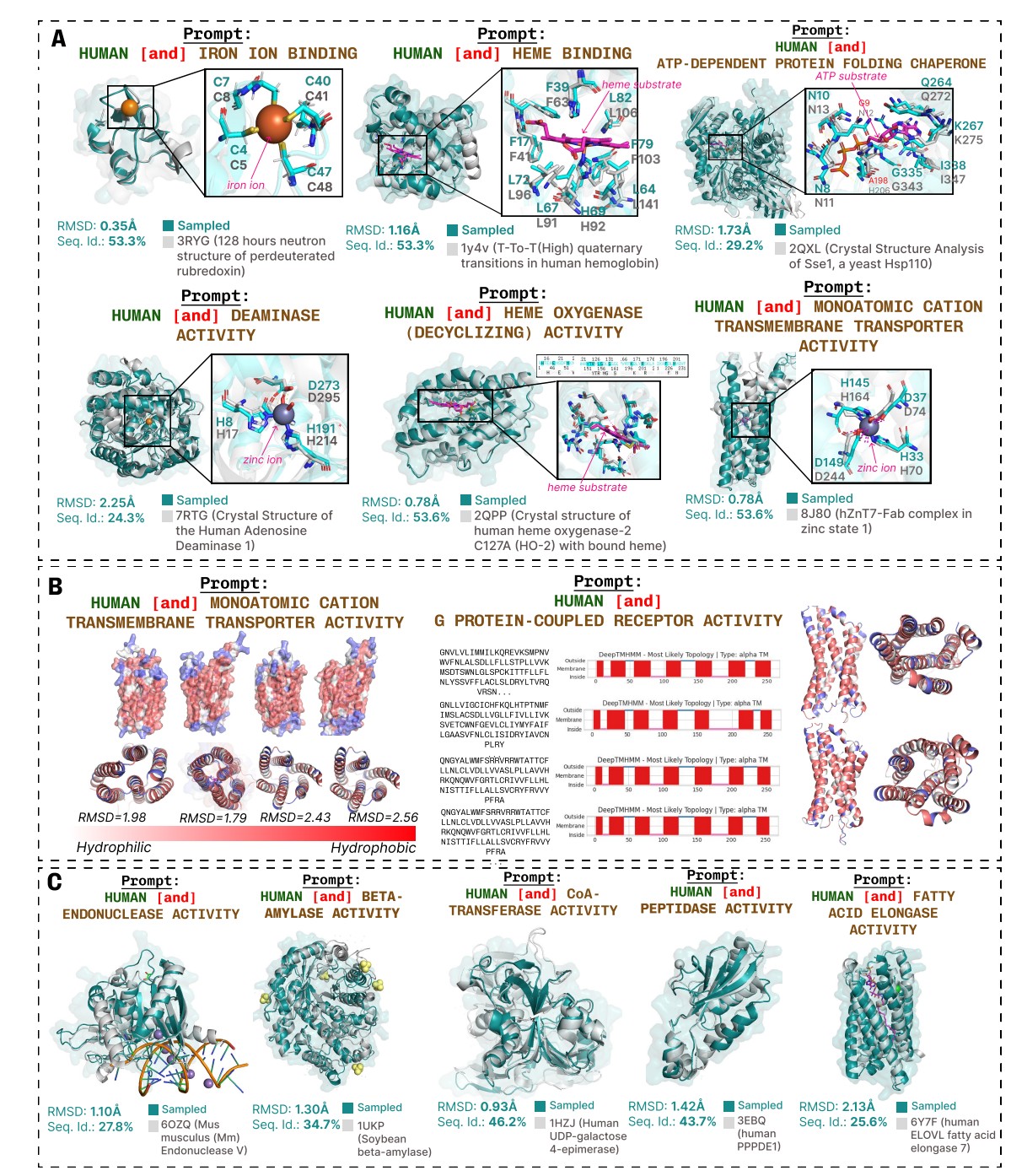

Figure 12: More case studies of conditional generations.

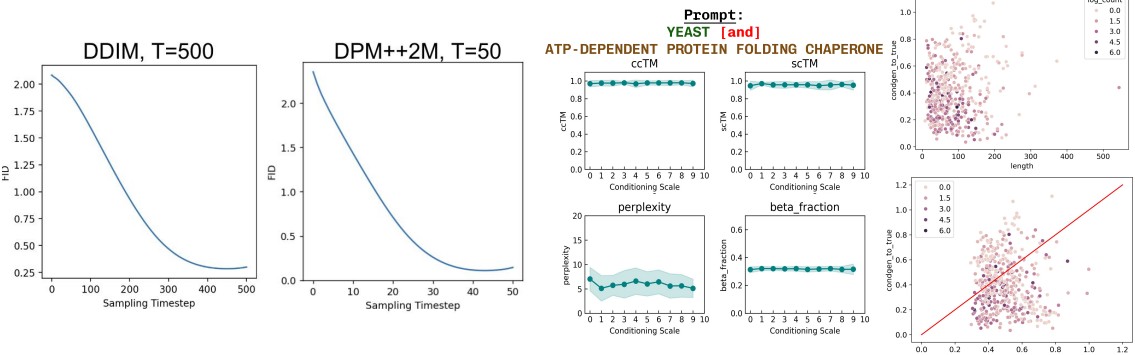

Figure 13: **(Left)** Frechet Distance between sampled protein and reference set of real protein, across sampling (reverse diffusion) timesteps, for the DDIM [33] sampler and the DPM++2M [52] sampler. For both, sample quality decreases steadily over time before plateauing. DPM++2M can achieve low FID results with only 10% of the original number of steps, but final results are still slightly worse. **(Center)** Examining the effect of conditioning scale on the output quality. **(Right)** Analyzing factors which may be contributing to a greater $\delta$ difference between the Sinkhorn distance of samples-to-real-functional-proteins vs random-proteins-to-real-functional-proteins.

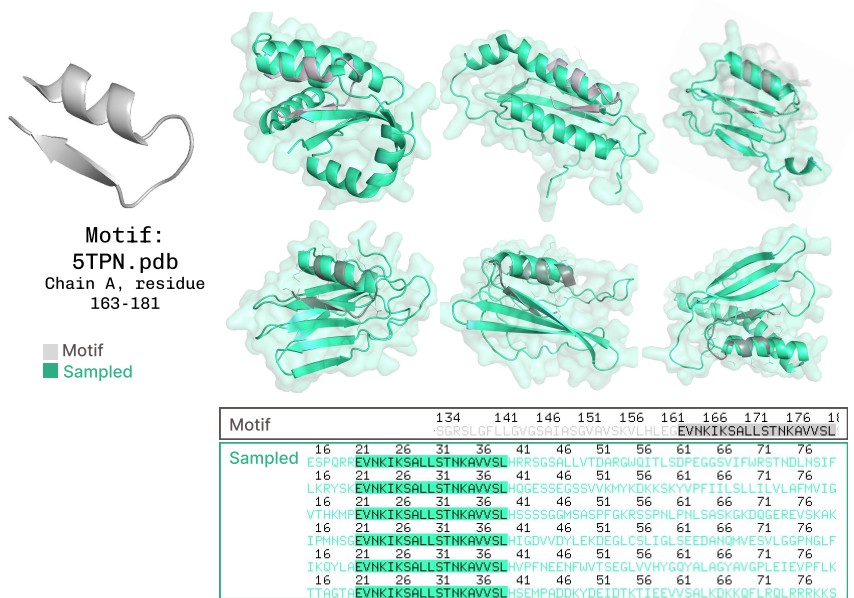

Figure 14: **Demo of motif scaffolding.** We use the same motif as the RFDiffusion [13] `design_motifscaffolding.sh` example for this experiment. The input motif is held constant at the user-prescribed location. Note that PLAID generates all-atom structure, whereas RFDiffusion does not position the sidechain atoms.

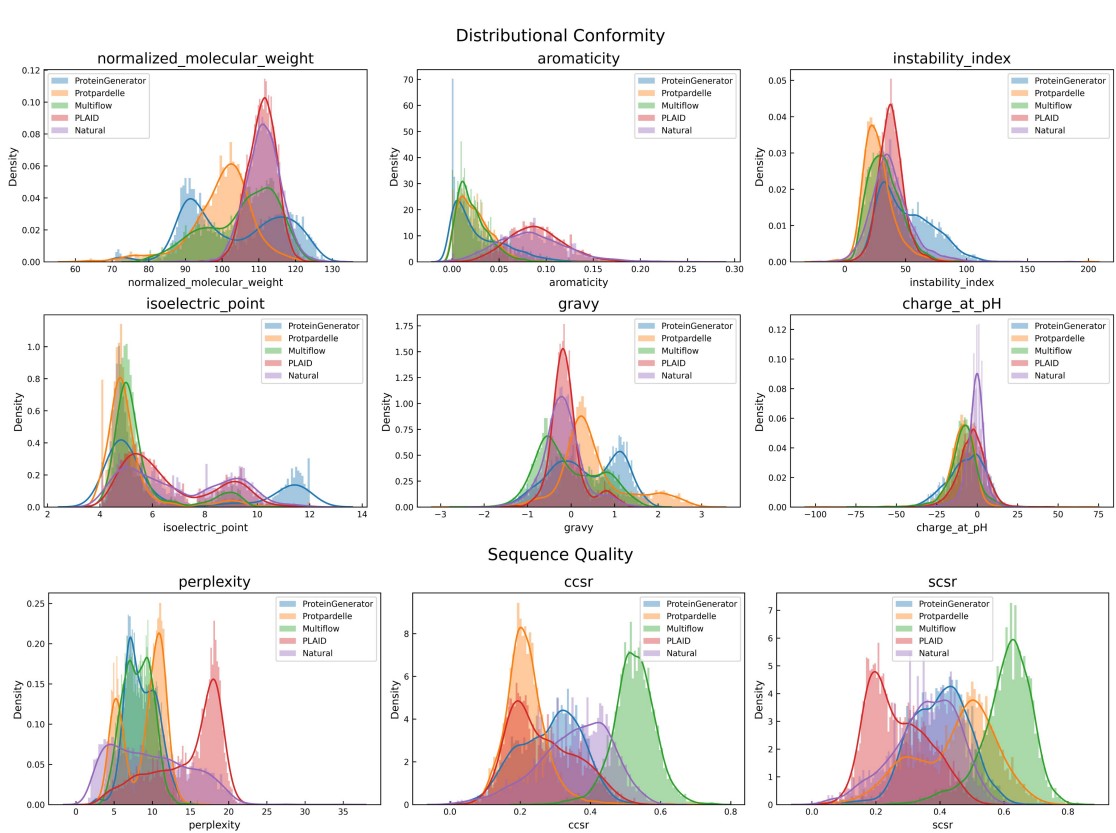

Figure 15: Examining histogram of metrics for nuanced comparison of how generated samples compare to that of natural proteins.

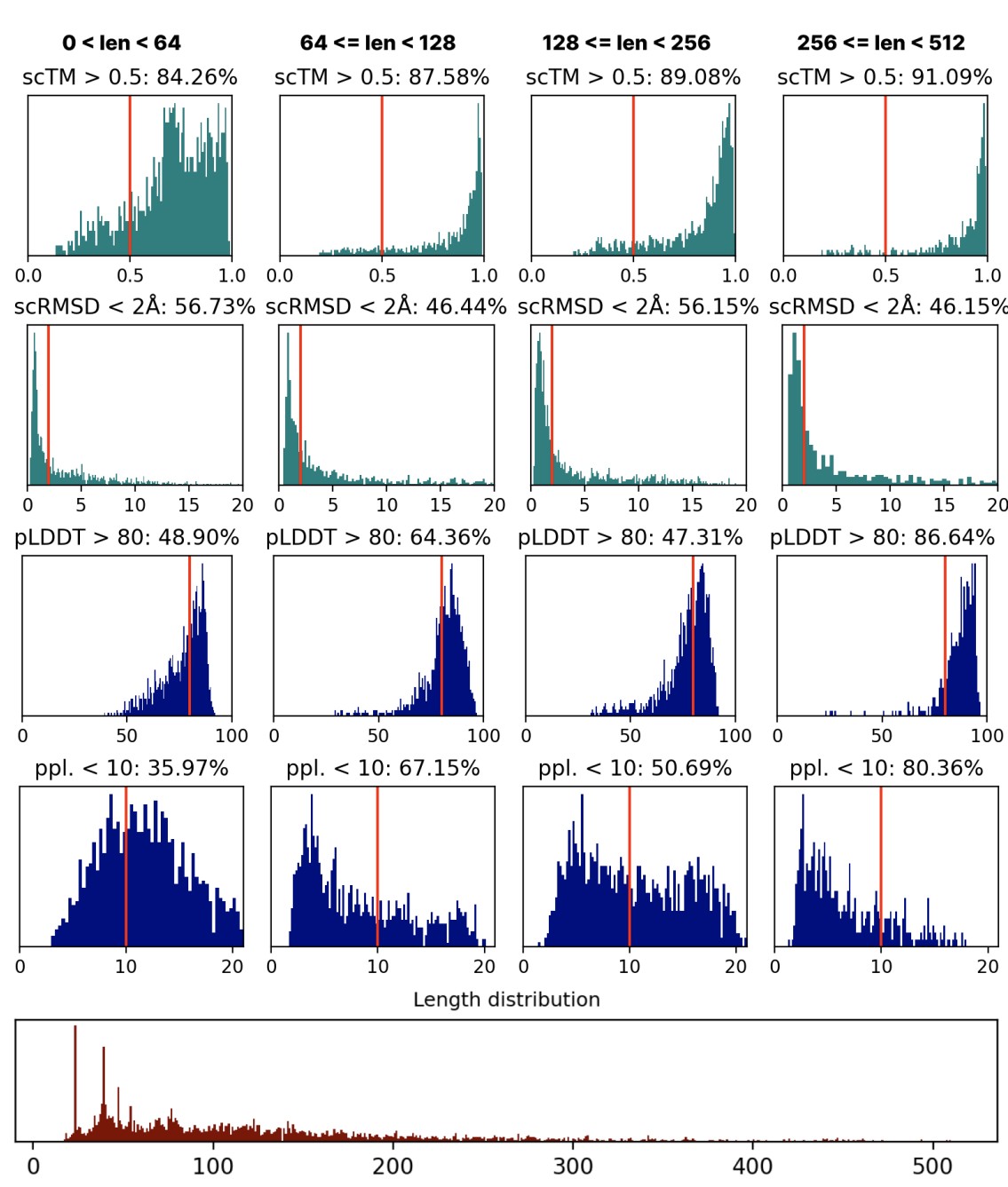

Figure 16: To examine the degree to which co-generation methods are overfitting to structure-based metrics, we examine properties on natural proteins.

