# OpenReview forum: "Generating All-Atom Protein Structure from Sequence-Only Training Data"
_ICLR.cc/2025/Conference — Submitted to ICLR 2025_

### Official Review · Reviewer_4icF · 2024-10-24

**Soundness:** 3
**Presentation:** 2
**Contribution:** 2
**Rating:** 6
**Confidence:** 3

**Summary:**

The manuscript proposed a latent-diffusion framework that can simultaneously generate protein sequence all-atom structure. The framework is called PLAID (Protein Latent Induced Diffusion). The framework leverages pre-trained protein representation and folding model (ESMFold) that generates latent representations containing both sequence and structure information of proteins. It also uses an autoencoder to further compress the latent representation for more efficient diffusion training. Such design makes training the model on larger sequence-only dataset possible. The framework also allows conditional generation.

**Strengths:**

1. The idea of compressing the learned representation from ESMFold and perform diffusion training in a more efficient manner is valuable, especially when generating larger protein is desired.
2. I generally agree with the authors that the representation from ESMFold contains both sequence and structure information, giving this work a rather solid foundation. The design in this work allows the model to be trained on larger sequence-only dataset is important.
3. Generating sequence and all-atom structure simultaneously is a relatively less-explored area. This paper is an interesting experiment.

**Weaknesses:**

1. The CHEAP autoencoder used in this work does not seem like a variational autoencoder. Therefore, the latent space may not be very smooth and can make diffusion model training difficult. I did not see an potential solution to that in the framework proposed by authors.
2. One advantage claimed by authors is that the framework allows the model to be trained on much larger sequence-only dataset. However, the unconditional generation performance of the model after training on Pfam is somewhat lackluster compared with MultiFlow. I also believe that an experiment showing how the performance of the proposed model scale with more data can be insightful.
3. The latent diffusion choice of the framework can be really beneficial in terms of speed when generating very long sequences (>600 aa). The authors did not show the performance of the model when generating very long proteins.

**Questions:**

1. Figure 2C is only a schematic of how CHEAP autoencoder is used in the framework, but not showing "without the normalization and compression post-processing steps in CHEAP, noise added in the latent space does not affect sequence and structure until the final timesteps in forward diffusion" as described in line 212-213. Is there another figure to visualize the effect of CHEAP autoencoder?
2. Conditional generation: how are conditions added to the model? Can authors elaborate or visualize?

---

> ### Author Response · Authors · 2024-11-24
> **Rebuttal by Authors**
>
> We’d like to thank the Reviewer for their insightful comments, and for highlighting the importance of being able to train on larger sequence-only datasets. We also are glad to see the Reviewer highlight the unique difficulties of all-atom generation.
>
> To address some weaknesses:
>
> _“The CHEAP autoencoder used in this work does not seem like a variational autoencoder. Therefore, the latent space may not be very smooth and can make diffusion model training difficult.”_
> * This is a great observation. In initial experiments, we struggled a lot with the ‘roughness’ of the inherent latent space of ESMFold; this is actually why we use the CHEAP autoencoder. CHEAP applies a channel normalization to combat massive activations, and though it is not an VAE, the authors show that the latent space is smooth and Gaussian-like after smoothing & compression.  If interested, Reference [1] discusses this at length.
> * We've **added a visualization of embedding values after compression in Appendix Figure 7** to address this comment.
> * We also **updated Figure 5** to show how noising in the latent space maps back to sequence and structure space before and after using the CHEAP encoder.
>
> _“The unconditional generation performance of the model after training on Pfam is somewhat lackluster compared with MultiFlow.”_
>
> * It's important to note that **Multiflow is not an all-atom generative model, and only generates backbones and the residue identity.** Please see our General Response for more details on this point.
> * When examining performance by length (**Updated Figure 5**), we also find that PLAID better balances sample quality and diversity at higher sequence lengths.
>
> _“I also believe that an experiment showing how the performance of the proposed model scale with more data can be insightful.”_
> * This is a wonderful suggestion. It was difficult for models to train for long enough to say something concrete about this before the end of the discussion period, but if accepted, we will add this to the camera-ready version.
>
> _“The latent diffusion choice of the framework can be really beneficial in terms of speed when generating very long sequences (>600 aa).”_
> * We thank the Reviewer for this suggestion. **We've added a comparison of sampling speeds for $L=600$ to Table 4**, both for the batched and unbatched setting.
>
>
> Re: CHEAP model details:
> * We’ve included more detail on how the CHEAP [1] module works. We've also added **experiments to show diffusion without CHEAP compression in updated Table 8**.
>
> Re: conditional generation details:
> * We’ve included a detailed panel **(Updated Figure 2D)** to describe how conditioning was added via AdaLN in DiT blocks. We also included a longer discussion of classifier-free guidance.
>
> **Conclusion.** We’ve aimed to address the Reviewer’s concerns around writing/figure clarity, performance, and included additional experiments for sampling speed. We hope the Reviewer will consider reviewing our updated manuscript and improve their score. We also welcome any additional comments.
>
> [1] Tokenized and Continuous Embedding Compressions of Protein Sequence and Structure. (Lu et al., 2024)

---

> > ### Comment · Reviewer_4icF · 2024-11-26
> >
> > I believe that the authors have solved some of my major concerns such as the smoothness of CHEAP encoder, long sequence generation, and comparison to MultiFlow. I do believe the quality of paper has improved as more evidences are provided. I would raise my rating. That being said, I do believe that the presentation can be further improved: Comparison between CHEAP and ESMFold latent space, data scaling etc. Minor suggestion would be to visually improve the manuscript by better aligning tables and figures with the margin etc. Looking forward to more polished manuscript.

---

> > > ### Author Response · Authors · 2024-11-26
> > >
> > > Thank you for raising the score to address the significant updates we've made in response to Reviewers' requests.
> > >
> > > On the comparison between CHEAP and ESMFold latent space, let us know how else we can clarify this. We've added:
> > > * **Figure 4**, which visually describes what happens when you add cosine noise to ESMFold latent space vs. CHEAP latent space
> > > * **Appendix Figure 9**, which illustrates embedding values
> > > * **Appendix Figure 10**, which provides the experiments for diffusing in the ESMFold latent space alone

---

### Official Review · Reviewer_WZZz · 2024-10-31

**Soundness:** 2
**Presentation:** 2
**Contribution:** 2
**Rating:** 5
**Confidence:** 4

**Summary:**

This paper introduces PLAID (Protein Latent Induced Diffusion), a diffusion model that can simultaneously generate protein sequence and all-atom structure, while only requiring sequence inputs during training. PLAID leverages the latent space of an existing protein structure prediction model, ESMFold, to capture the joint distribution of sequence and structure. By defining the training data distribution based on sequence databases rather than structural databases, PLAID can access a much larger and more diverse set of protein data, increasing the available annotations and enabling controllable generation along axes like protein function and organism of origin. PLAID avoids the need for alternating between sequence and structure generation steps, and can directly sample from the joint distribution of sequence and all-atom structure.

**Strengths:**

- Leveraging the latent space of existing structure prediction models for protein structure (all-atom) and sequence co-design is a refreshing and innovative idea.
- The paper presents a relatively clear discussion of the main components of the methodology.

**Weaknesses:**

- My main concern with the paper lies in its underwhelming performance. While I generally believe the approach is feasible, the current experimental results suggest that more effort is needed to thoroughly explore effective strategies for co-design within the latent space of structural prediction models.
- The lack of support for motif scaffolding raises concerns about the controllability of the current model. Although function and organism terms can be used as conditioning, this form of conditioning may still fall short in achieving precise control.
- The paper lacks a comprehensive discussion of related work. For instance, as a method for protein design utilizing structure prediction models, it overlooks hallucination models [1] that also manipulate structure prediction models in similar ways.
- Several important details critical for understanding the paper are not explained:
   - There is no explanation provided on how the model integrates function and organism conditioning. The related model design strategies require further clarification.
   - The design of the sequence decoder $\phi^{-1}_{\text{ESM}}$ also requires further elaboration.
   - The paper does not provide an explanation of how FIDs, originally designed as a metric for image generation, is adapted for application in protein design.
- There are numerous errors in the citations of images and text within the paper:
   - The reference to Figure 2 between lines 188 and 216 does not align with the textual description and needs to be corrected.
   - The description of the probabilistic decomposition for different methods in line 173 contains errors.
   - The yellow and green dots in Figure 4D lack a legend explanation.

[1] Anishchenko I, Pellock S J, Chidyausiku T M, et al. De novo protein design by deep network hallucination[J]. Nature, 2021, 600(7889): 547-552.

**Questions:**

- If we focus less on the structures obtained from experiments and instead accept those predicted by models, the available structural data is still substantial and does not exhibit the vast gap claimed in the paper. In fact, many current studies already incorporate predicted structures in their training processes. Therefore, the paper's statements may not accurately reflect the current reality.
- Does designing based on structure prediction models introduce an unnatural bias toward certain protein structures? For example, AF2 has been found to predict overly "clean" structures, often lacking the unstructured regions that are crucial for capturing protein dynamics.
- Is there a more detailed explanation or experimental validation available for using the CHEAP module?
- Why is a design result that finds more homologous sequences considered more novel in the context of defining Hit metrics?

---

> ### Author Response · Authors · 2024-11-24
> **Rebuttal by Authors**
>
> We want to thank the Reviewer for acknowledging the innovative and refreshing aspects of our approach. We're also grateful for the suggestions, which have helped us make many improvements. We'd appreciate it if you could take a new look at the PDF file and see if your concerns were addressed.
>
> To address the stated weaknesses:
>
> **Re: performance:** _“My main concern with the paper lies in its underwhelming performance…”_
> * We've greatly expanded the depth with which we've been examining the methods. When we expanded sample lengths to 512 and examined performance by length, we found that PLAID is **better at balancing quality and diversity for longer sequences (Updated Figure 6 & Appendix Figure 11)**.
> * It should be emphasized the **Multiflow is not an all-atom method. Amongst all-atom generation methods, PLAID achieves SOTA on consistency and naturalness metrics**, both of which are important metrics that have been neglected by co-generation literature. Please see our General Rebuttal for more on this point.
>
> _“The lack of support for motif scaffolding raises concerns about the controllability of the current model. Although function and organism terms can be used as conditioning, this form of conditioning may still fall short in achieving precise control.”_
> * We were deliberate in choosing capabilities that would enlarge the surface area of how generative models can help drug development. **Organism conditioning can be key to many developability and humanization bottlenecks. GO terms act as proof-of-concept for the eventual aim of scaling to more complex annotations, such as natural language**, which affords precise control that is relevant to many domains. Please see our General Response for additional detail on this point.
> * In our updated manuscript, to highlight precise control, we’ve investigated function-conditioned sampling, and found that **PLAID preserves precise catalytic motifs (Updated Figure 7, Appendix Figure 12).**
>
> _“The paper…overlooks hallucination models that also manipulate structure prediction models in similar ways.”_
> * We thank the reviewer for feedback on our Related Works section and have expanded the discussion to address how our approach differs substantially from hallucination. Anishchenko et al., focus on sequence design with MCMC in sequence space. It doesn’t propose new structures, and instead looks for a sequence that folds into that model. In contrast, we are doing multimodal / all-atom design in latent space. This problem setting encompasses that examined in Anishchenko et al., but is vastly more difficult, since we also search through structure space.
>
> _“The related model design strategies require further clarification...”_
> * Thank you for this feedback. We've added an **ablations table (Updated Table 1)** to highlight this. We've also rewritten the Methods section. Many design decisions were driven by **scalability and generalizability** to new methods, and being able to use maximally efficient attention kernels, such that it would be easier to build upon it (either the method or the weights itself).
>
> _“There is no explanation provided on how the model integrates function and organism conditioning…”_
> * We added a detailed panel **(Updated Figure 3D)** which describes the AdaLN operation used in DiT blocks to incorporate conditioning information.
>
> _“The design of the sequence decoder…also requires further elaboration…”_
> * As noted in Methods, the sequence decoder is taken from [1]. However, we've incorporated the feedback and restructured our manuscript to make this information easier to find.
>
> _“The paper does not provide an explanation of how FIDs…is adapted for application in protein design.”_
> * This is a good point, and in retrospect, “Frechet Distance” would have been a better description. In our **Updated Figure 8**, we instead use Sinkhorn Distance between the generated latents and the corresponding embeddings from a distribution of real proteins. This also allows us to use smaller sample sets, so we are able to compare more classes.
>
> (Continued below...)

---

> ### Author Response · Authors · 2024-11-24
> **Rebuttal by Authors (Continued)**
>
> To address questions:
>
> _“If we focus less on the structures obtained from experiments and instead accept those predicted by models, the available structural data is still substantial.”_
> * Synthetic data has been known to create errors such as mode collapse [2]. (Interestingly, this is precisely what we observe in Multiflow for higher sequence lengths, which does use distillation.)
> * Using latent diffusion in a compressed space allows easier exploitation of hardware-aware attention kernels. Structure inputs often require architectures and data representations less amenable to scaling.
> * Our approach democratizes all-atom generation. Synthetic data is computationally expensive to generate, but PLAID can be used as long as model weights are available from a pretrained foundation model.
>
>
> _“Does designing based on structure prediction models introduce an unnatural bias toward certain protein structures? For example, AF2 has been found to predict overly "clean" structures.”_
> * We agree that the bias towards overly “clean” structures is a pervasive issue, and this actually is precisely what motivated our work. The inherent bias towards “clean” and crystallizable structures in the PDB is severe; by incorporating data from sequence databases, we can actually sample from those regions, which would otherwise not be captured in existing protein structure generation methods.
>
> _“Is there a more detailed explanation or experimental validation available for using the CHEAP module?”_
> * Thanks for this suggestion; we’ve **added results from learning on the latent space without using the CHEAP module (Updated Appendix Figure 8)**, and also plot & visualize how noise in the latent space maps back to corruptions in sequence and structure space **(Updated Figure 4)**. We also visualize the latent space in **Updated Appendix Figure 7**.
>
> _“Why is a design result that finds more homologous sequences considered more novel in the context of defining Hit metrics?”_
> * We've removed Hit % as it was a confusing choice of metrics. In **Updated Table 3**, we use sequence identity % to the closest mmseqs neighbor and structure TMScore to closest foldseek neighbor, to assess sequence and structure novelty, respectively.
>
> **Conclusion.** We thank the Reviewer for their detailed examination and suggestions, which has improved this work. We hope that our response make our reasoning around choice of conditioning clearer, and that our updated results reflect how PLAID circumvents mode collapse observed at longer lengths in other approaches. We would ask the reviewer to consider increasing their score, and are happy to address any additional comments or suggestions during the discussion period.
>
>
> [1] Tokenized and Continuous Embedding Compressions of Protein Sequence and Structure. (Lu et al., 2024)
>
> [2] Synthetic Data, Real Errors: How (Not) to Publish and Use Synthetic Data. (Breugel et al., 2023)

---

> ### Comment · Reviewer_WZZz · 2024-11-27
>
> The author's response has addressed some of my concerns, and I have accordingly raised my score. However, I still feel that the author’s perspective on their work is too narrowly defined. Specifically, whether one designs structure and sequence simultaneously, designs structure first and then sequence, or explores the sequence space before folding the structure, all fall within the realm of protein design. A comprehensive study should include comparisons of different approaches.

---

> > ### Author Response · Authors · 2024-11-27
> >
> > We appreciate you engaging with our work and improving the score.
> >
> > With respect to comparisons to generating sequence/structure independently: this actually is reflected in our self-consistency scores in Table 2. scTM and scRMSD is generating structure first, then decoding to sequence. scSR is generating sequence first, then turning to structure:
> >
> > | Model | Structure Quality |  |  | Sequence Quality |
> > |------------------|------------------|------------------|------------------|------------------|
> > | | scTM (↑) | pLDDT (↑) | Beta sheet % (↑) | scSR (↑) |
> > | ProteinGenerator | 0.72 | 69.00 | 0.04 | 0.40 |
> > | Protpardelle | 0.57 | N/A | 0.11 | 0.44 |
> > | PLAID | 0.64 | 59.46 | 0.13 | 0.27 |
> > | Multiflow* | 0.91* | N/A | 0.10* | 0.61* |
> > | Natural | 0.84 | 84.51 | 0.13 | 0.39 |
> >
> > We've also compared to ProteinGenerator and Protpardelle, which are also models that generate one first before the other. PLAID achieves better scTM scores than Protpardelle, which uses ProteinMPNN. scSR is fairly low across the board, including for natural proteins.
> >
> > We also want to note that **we've added a motif scaffolding experiment in Figure 14**, to address the Reviewer's concerns in the original comment.
> >
> > Would these experiments altogether address the Reviewer's remaining concerns from accepting this paper? If so, we'd be grateful if you can consider reflecting this in your score, or provide further feedback.

---

### Official Review · Reviewer_n4hb · 2024-11-02

**Soundness:** 2
**Presentation:** 2
**Contribution:** 2
**Rating:** 6
**Confidence:** 2

**Summary:**

The paper provides a way to generate all atom protein structure using only sequence data during the training without requiring intermediate structural inputs.

**Strengths:**

The training does not require sequence data. By training on sequence data alone, PLAID can scale across larger datasets compared to models constrained by experimentally resolved structures.

**Weaknesses:**

1. The model’s reliance on ESMFold’s latent space and pre-trained weights, while advantageous, could be limiting, and main factor contributing the limited performance.
2) The paper can benefit from improved writing, I think some parts of the paper need more explanation.

**Questions:**

1) The statement in line 178, p(x) = \phi(s) = \phi(\omega), I am not sure if this is correct, p(x) supposed to be the joint while the other two is individual distributions.
2) how the ESMFold first component that maps the sequence to the latent captures evolutionary prior? and what is the intuition behind that prior?
3) The equation at the end of the line 187, isn't the structure module takes actually x and map it to the strucuture \omega, the equation written as if it takes the structure as input?
4) line 195, f_ESM is not defined. I am not sure if I understood the part the sequence decoder need to be separately trained? The block B ( in figure 2) is used off the shelf or it is also trained?
5) the line 199 ( the inference ... is shown in Figure 2B, is it typo? the inference is shown in D in figure 2?
6) if I summarize the training of this model, one need a pre-train ESM2 model and CHEAP, then during training, the sequence of protein go through ESM2, mapped to x, then (not sure how the x_norm is obtained), x_norm goes through CHEAP encoder, gives us more compressed rep x_0 and there we train a diffusion model , So only trainable params are the diffusion model in the space of x_0,  and inference time,  we use the frozen CHEAP decoder and ESM structure and sequence decoder, is that right what I summarize?
7) Any intuition behind the why the compressed latent first dimension set L/2?
8) I am wondering what happens if one apply the diffusion model directly on the latent of the ESM2 model, I think the line between 211 and 215 tried to answer that but not sure if I understood.
9) The figure 4D has beed resolution, and  color of the round dots are a bit misleading.
10) what is the red vertical line in the figure 4 B represents?
11) I am wondering what happens if one trains the full Model in figure 2C instead of freezing some parts?

---

> ### Author Response · Authors · 2024-11-24
> **Rebuttal by Authors**
>
> We thank the reviewer for these helpful comments, and for acknowledging the method’s strength in being able to scale across larger datasets.
>
> In Weaknesses, the Reviewer makes an astute observation that the performance of ESMFold limits this model. We think an extension of this paradigm to include a finetuning of the ESMFold model, or developing new frontier biological foundation models for PLAID, would be extremely useful – hence why we’d like to share this work with the research community.
>
> Regarding questions:
>
> * _“The statement in line 178, p(x) = \phi(s) = \phi(\omega), I am not sure if this is correct…”_
>
> $\phi(\cdot)$ is a mapping that transforms different distributions. We’ve updated the writing to make this clearer.
>
> * _“how the ESMFold first component that maps the sequence to the latent captures evolutionary prior?”_
>
> ESMFold uses ESM2 in the first component, which was trained with a masked language modeling objective on UniRef. This is also empirically justified by the fact that replacing the explicit MSA construction in AlphaFold with ESM2 yields comparable results. We’ve updated the writing to clarify this.
>
> * _“The equation at the end of the line 187, isn't the structure module takes actually x and map it to the strucuture \omega, the equation written as if it takes the structure as input?”_
>
> We use the notation for inverting a function to say that an inversion of the structure module would provide a mapping from structure to $x$.
>
> * _“I am not sure if I understood the part the sequence decoder need to be separately trained”._
>
> Yes; as noted in the Methods, the sequence decoder is separately trained, and provided in [1]; it reaches a validation accuracy of 99.7%. We’ve reorganized the sections to make this easier to find.
>
> * _“the line 199 ( the inference ... is shown in Figure 2B, is it typo?”_
>
> Thank you for noticing this. We’ve entirely updated Figure 2 to also include a schematic of how conditioning information is incorporated into each DiT block.
>
> * _“if I summarize the training of this model, …”_ Yes, this is the correct summary. _“...not sure how x_norm is obtained…”_
>
> This comes from the CHEAP [1] model, where massive activations are removed via a post-hoc channel norm operation. We've added a longer discussion of details of the CHEAP model.
>
> * _“Any intuition behind the why the compressed latent first dimension set L/2?”_
>
> This reduces memory usage, since we use a transformer architecture for scalability, and memory increases quadratically with length. We added this clarification to the updated manuscript.
>
> * _“I am wondering what happens if one apply the diffusion model directly on the latent of the ESM2 model…”_
>
> We’ve added **Appendix Figure 8** to show performance before and after compression, and include a longer discussion of high-resolution image synthesis via latent diffusion in related works.
>
> * _“The figure 4D has beed resolution, and color of the round dots are a bit misleading.”_
>
> We’ve incorporated this feedback and updated this figure with expanded results in **Updated Figure 9**.
>
> * _“what is the red vertical line in the figure 4 B represents?”_
>
> This is the threshold for which “designability” is defined. In the **Updated Figure 6**, which examines performance by length, we’ve made sure to make this more clear.
>
> * _“I am wondering what happens if one trains the full Model in figure 2C instead of freezing some parts?”_
> This is a great suggestion and opportunity for follow-up work. It should presumably improve performance, but retraining/finetuning ESMFold in addition to the models trained for this paper was not feasible with our resource constraints.
>
> **Conclusion.** We're very grateful for the comments from Reviewer n5hb, and hope that the updates can address these concerns. It would be great if the Reviewer could improve their score, and we’re happy to address any further comments before the end of the Discussion Period.

---

> ### Author Response · Authors · 2024-12-04
>
> As the discussion period is coming to a close, we'd appreciate it if Reviewer n4hb can find a chance to review the updated PDF. We've made significant changes to address the Reviewers' comments, including **rewriting the Methods section to reflect Reviewer n4hb's suggestions**, experiments that analyze when and how PLAID outperform baselines, expanded discussion of why we used embedding compression, and better highlighting how PLAID correctly positions sidechains at active sites when prompted by function.
>
> If we've been able to address the Reviewer's concerns on the impact of this paper, we'd appreciate it if you can improve your score accordingly. Thanks for taking time to engage with our paper.

---

### Official Review · Reviewer_bWWc · 2024-11-02

**Soundness:** 2
**Presentation:** 2
**Contribution:** 3
**Rating:** 5
**Confidence:** 4

**Summary:**

The authors propose a new method for generating all atom protein structures from sequence-only training data which they call PLAID (Protein Latent Induced Diffusion). For this, they leverage the pretrained ESMFold latent space as well as the compressed CHEAP embeddings and train a Diffusion Transformer over these CHEAP embeddings and also create a sequence decoder from the ESMFold latent space. With that, at inference time they generate first a CHEAP embedding, decode that to an ESMFold embedding and then use the ESMFold structure decoder as well as the newly trained sequence decoder to recover structure and sequence. Via classifier-free guidance they allow conditional generation with GO ID and organism labels and compare the performance of their method to other co-generation methods and to reference datasets from the PDB.

**Strengths:**

1. **Novelty of approach**: While latent diffusion models have been described before, this is the first work that describes the usage of a latent diffusion model over protein language model embeddings to generate both sequences and all-atom structures. This makes dealing with varying lengths of side chains easier than in previous all-atom generation methods like Protpardelle and allows leveraging capabilities from pre-trained models like ESMFold.
2. **Leveraging compute and data scale**: The compressed latent space as well as the standard Diffusion Transformer architecture allow the model to train and generate efficiently. The fact that only sequence data is used for the actual diffusion model training also allows scaling to larger datasets as well as leveraging potentially more diverse training data. It also allows the integration of sequence labels as conditioning information as presented via GO IDs.

**Weaknesses:**

1. **Designability/Diversity Performance**: The model underperforms quite strongly on Cross-Modal consistency and diversity compared to Multiflow. Especially the Co-Designability number of 40% shows that the model is not performing well on the intended task of protein structure and sequence generation compared to work that is out there such as Multiflow.
2. **Novelty Performance**: It is not really clear why the authors bold their novelty Hit% number, implying that their model performs best on novelty. Multiflow outperforms on 1-TM score (but its number is not bolded), and if I understand correctly Hit% numbers indicate how often homologs can be found in a sequence database; a higher value here does not imply that the model is producing more novel samples, if anything it is the opposite. Also in the SeqID% column, it is not clear why the 83.6% number of the natural reference set is bolded since the authors claim a lower number on these metrics is better and the natural reference set is just a reference set and not a baseline model. It would be helpful if the authors review the table again and include consistent bolding to make the information in that table more accessible to the reader. It would also be beneficial to describe in more detail why 1-TM score and Hit% are useful metrics and how higher/lower values reflect the performance of the evaluated models.
3. **Missing control experiment for claims**: The PLAID model is only evaluated via cross-modal consistency, i.e. refolding accuracy using the sequence generated from the model. However, past publications have shown that in many cases the sequence generated from the model yields lower self-consistency values than a sequence generated from the generated structure via ProteinMPNN. To truly evaluate whether their model has cross-modal consistency, the authors should compare to this baseline and show that their consistency is higher than when they use ProteinMPNN (see the MultiFlow paper for experiments that show these kind of results).
4. **Naturalness of sequences**: The authors claim that their model can generate more natural sequences than other models. First of all, it is not clear why the molecular weights are a lot higher for all methods compared to the natural reference set? I would assume that to compare fairly one should sample proteins with similar lengths compared to the reference set, but this does not seem to be the case here. And while the model has values that are more similar to natural proteins in terms of isoelectric point, gravy and charges (charge at pH does not note which pH is used for the calculation here?), the samples seem to be a lot more instable compared to MultiFlow's and the baseline samples, which seems one of the most important properties in practical applications.
5. **Mistakes/missing figures**: There seem to be several typos/mistakes in the paper that makes following the flow difficult to understand, for example
   1. L173: Protpardelle and ProteinGenerator have the same factorization although the one for ProteinGenerator should be switched around.
   2. L191: Diffusion training is depicted in Fig 2C and not 2A.
   3. L199: All-atom sampling is depicted in Fig 2D and not 2B.
   4. L212: "Figure 2C shows that without the normalization and compression post-processing steps in CHEAP, noise added in the latent space does not affect sequence and structure until the final timesteps in forward diffusion". Figure 2C does not show this, it is just a illustrative graphic of the training process. The mentioned content regarding noise not affecting sequence and structure until final timesteps is not mentioned anywhere else and seems to be missing from the paper. It would be helpful if the authors conduct a thorough review of all figure references and ensure that all mentioned content is actually included in the paper. This would help improve the overall clarity and coherence of the manuscript.
   5. L214: "(SNR and log-SNR curves shown below)". There are no SNR and log-SNR curves to be found in the paper.
6. **GO term FID score**: The authors claim that their model can generate realistic proteins with a given GO ID and measure this by an FID score. However, FID score (Frechet Inception Distance) is a metric for judging image quality for conditional image **generation** tasks where Inception refers to the model used to embed the images and compare the embeddings against the reference set. Since using the Inception model in this context does not make sense for proteins, the FID score as presented here does not make much sense. If a different model is used in this paper for embedding and comparing these embeddings against a reference set, the authors should mention the model that was used there, the reference set that was used and show validation experiments to demonstrate that their proposed new metric (which would not be an FID score anymore) has any relevance/utility in the protein domain. It would be helpful if the authors can provide a detailed explanation of how they adapted the FID score for proteins, including specifics on the embedding model used and the reference set. Additionally, validation experiments or justification for using this metric in the protein domain would strengthen the evaluation approach.

**Questions:**

1. In Table 1, the authors imply that their model can perform two GO term and organism conditioning, while the other methods according to the table cannot do conditioning, even though for example RFDiffusion can do a lot of conditioning constraints that PLAID cannot (symmetric oligmers, binders, ...) which for practical applications have arguable more relevance. Did the authors explore these other conditioning approaches and could the table be updated accordingly?
2. Since the model is trained on protein domains in Pfam only, does this limit the generation capabilities to single-domain proteins?
3. In newer work like ESM3, the language model is directly used for structure generation via explicit structure token embeddings. How does this compare to the approach proposed in the paper and what are potential advantages and disadvantages?
4. It is stressed in the paper that the model can sample more natural sequences than other models. Besides questions about this claim itself (see weaknesses section), should naturaleness in terms of aromaticity etc be a target for a "good" protein generative model? For many applications these models are used for non-natural properties like higher melting temperatures etc might be very useful.
5. The authors show case studies for conditional generation with GO IDs. But while they show low sequence identiity, is there any way of judging how similar the global structure is, not just the sequence? In addition, how well do these case studies work across the board beyond the two examples shown in the paper?
6. In protein generation it is often useful to employ structure constraints such as motifs, specific secondary structure or binding interfaces for generating samples. Is this possible in a latent framework such as the one proposed here?

---

> ### Author Response · Authors · 2024-11-24
> **Rebuttal by Authors**
>
> We thank Reviewer bWWC for the insightful comments that have helped us improve this work, and for acknowledging the novelty and scalability of our approach.
>
> To address the stated weaknesses:
> 1. **Designability/diversity performance**:
> * We'd like to note that **Multiflow does not produce side chain positions**; if we assume each side chain to have 0 to 4 rotamers, this induces $4^L$ more degrees of freedom. Multiflow also cannot mirror PLAID’s ability to capture sidechain placements and precisely model ion binding and active sites **(Updated Figure 6)**.
> * Designability metrics are extremely fragile, and have strong structure bias. Since ProteinMPNN, etc. are trained on the PDB, these metrics essentially assess if new samples are in distribution with the PDB. Our goal is explicitly to remove over-emphasis on structure. They are also fallible; the reported sequence recovery rate of ProteinMPNN is 52.4% [2].
> * Natural proteins themselves have low self-consistency performance. This can be seen in Tables 2 & 3. This is also why we emphasize cross-consistency; as seen in **Updated Table 2**, natural proteins actually achieve near perfect performance here.
> * Novelty performance: in our **Updated Table 3**, we’ve removed names like “Hit %” that have caused confusion.
>
> 2. **Control experiment using ProteinMPNN:**
> * We’ve added this experiment by using ProteinMPNN to predict sequence from a given structure, and compare its performance to the original structure (i.e. the scTM and scRMSD metrics) in Updated Figure 5, Table 2, 3. Consistency here is lower than using the sequences directly produced by PLAID. This reaffirms our original thesis and the simplicity, robustness, and novelty of end-to-end training and co-generation without alternating model calls.
>
> 3. **Naturalness of sequences**:
> * We’re grateful for the Reviewer’s careful consideration and have updated the table to reflect **length normalized molecular weights**.
> * Re: stability metric, this is a heuristic calculated from the prevalence of dipeptides, and should not be interpreted as a gold standard. * To illustrate this, we’ve included a stability distogram of real, experimentally stable proteins in Updated Appendix Figure 10E.
>
> 4. **Ambiguous usage of FID term**:
> * We apologize for confusion here; “Frechet Distance” would have been more suitable. We updated this experiment using Sinkhorn Distance. This is also more resilient to small sample sizes, which enabled us to look at more function classes.
> * As to why we run this experiment: this assesses the distance in latent space between the real and generated proteins, to assess “proteinness” separately from decoder performance.
>
> Regarding questions:
> 1. **Conditioning capabilities**:
> _In Table 1, the authors imply that their model can perform two GO term and organism conditioning, while the other methods according to the table cannot do conditioning…_
>
> We’ve removed Table 1 given that it’s impossible to do a full comparison.
>
> _“...for example RFDiffusion can do a lot of conditioning constraints that PLAID cannot (symmetric oligmers, binders, ...) which for practical applications have arguable more relevance.”:_
>
> We’d respectfully disagree that function & organism conditioning are less relevant; they are in fact crucial to developability and humanization bottlenecks in drug discovery, and introducing new capabilities should be considered a strength. Please see our General Rebuttal for more on this point.
>
> 2. **Re: domain generation:**
> _“Since the model is trained on protein domains in Pfam only, does this limit the generation capabilities to single-domain proteins?”_
>
> We discuss our rationale for using Pfam in the “data” section. In brief: (1) memory scales quadratically with length, so we wanted to maximize content in short sequences; (2) It’s difficult to perform in silico comparisons on full proteins containing disordered regions; (3) Most of our evals are around sequence-structure and motif conservation, so nothing is lost by focusing on single domains.
>
> (continued below)

---

> ### Author Response · Authors · 2024-11-24
> **Rebuttal by Authors (Continued)**
>
> 3. **Re: similarities to ESM3:**
> _“In newer work like ESM3, the language model is directly used for structure generation via explicit structure token embeddings. How does this compare…”_
> * This is an interesting insight. This work was developed concurrently with ESM3; we’d also considered a unified tokenized approach at first, but chose diffusion because existing literature shows that diffusion models at similar parameter counts can achieve better performance than autoregressive ones. [4]
> * It’s quite straightforward to extend PLAID to use tokenized representations, since CHEAP [2] embeddings also include a tokenized version.
> * We've updated the related works based on this suggesiton.
>
> 4. **Re: naturalness:**
> _“…should naturalness in terms of aromaticity etc be a target for a "good" protein generative model?”_
>
> Previous literature [1] shows that distribution conformity to natural proteins is very correlated with expression. (Please see our General Response for further discussion.)
>
> _“...For many applications these models are used for non-natural properties like higher melting temperatures etc”_
>
> This is a fascinating suggestion. In PLAID, we can do this by conditioning by a  thermophilic organism. We'll add this experiment before the end of the discussion period.
>
> **Re: conditional generation evaluation:**
> _“...how well do these case studies work across the board beyond the two examples shown in the paper?”_
>
> We have significantly expanded our analyses, **showing that function-conditioned generations conserve active site residues, global structural similarity, and hydrophobicity patterns of membrane proteins.**
>
> **Re: PLAID compatibility to other conditioning capabilities:**
>
> _“In protein generation it is often useful to employ structure constraints such as motifs, specific secondary structure or binding interfaces for generating samples. Is this possible in a latent framework such as the one proposed here?”_
> * To guide PLAID by per-token secondary structure, one can use a text-conditioned approach. In an early iteration of the model, we’d conditioned the model by secondary structure content, but are focusing on GO term and organism control for better applicability to drug development and introducing new capabilities for all-atom generative models.
> * For binding interfaces, one can take an in-painting approach. Since at inference time, the user specifies the length, one can provide a sequence and leave “additional space” for the model to in-fill the binder.
> * For motif scaffolding, one can keep the input motif fixed during inference.
>
> **Conclusion.** The Reviewer’s suggestions have been helpful for improving the clarity of our paper and the strength of our claims. In light of the additional results we provide, we’d appreciate it if the Reviewer would consider reassessing this work and improving their score.
>
> [1] Protein Discovery with Discrete Walk-Jump Sampling. (Frey et al., 2023)
>
> [2] Tokenized and Continuous Embedding Compressions of Protein Sequence and Structure. (Lu et al., 2024)
>
> [3] GLIDE: Towards Photorealistic Image Generation and Editing with Text-Guided Diffusion Models. (Nichol et al, 2021)
>
> [4] Denoising Diffusion Probabilistic Models. (Ho et al., 2021)

---

> ### Comment · Reviewer_bWWc · 2024-11-26
>
> I appreciate the effort the authors put into the rebuttal by fixing the mistakes and missing figures in the manuscript, clarifying some of my questions regarding naming and metrics in the paper and conducting additional experiments and increase my score accordingly. Some mistakes such as missing reference links (see "Table ??" at the end of page 8) and should be fixed.
>
> I think the paper has an interesting novel approach as mentioned before and can be turned into great work. However, as I mentioned in my initial review, a few things such as some of the experiments now acknowledged as interesting by the authors as well as proper metrics validating the approach should be added in order to make the work more convincing and stronger. Some more detailed responses below.
>
> 1.
> - Although I agree that designability metrics etc can be biased for structure, it is an experimentally verified metric for lab success of designed proteins. It is not necessarily the aim of protein design to exactly mimic all natural protein properties (e.g. better expressibility, thermostability etc). If the authors think that their approach would benefit from a different metric, they should compare on that metric and thoroughly verify that this is a meaningful metric in a practical sense.
>  - Re the point about the MultiFlow comparison: it is not fully clear to me yet why the all-atom capability in your use cases is necessarily improving upon just modelling backbones + sequence as in Multiflow; experiments that show the advantage of this capability via side-chain conditioning as in other recent work [1] or other leverage of the all-atom capacity for design would strengthen that point. As long as such things are not possible, the clear advantage of producing an all-atom structure in the end is not obvious to me.
> - What is meant regarding ProteinMPNN being fallible with a 52% sequence recovery? Of course, the model is not perfect, but why is the 52% sequence recovery indicative of that? I at least could not point to the "best" value for sequence recovery since many sequences can fold into the same structure.
>
>
> 2. FID vs Sinkhorn: I still do not understand what model was used for the Frechet distances; this needs to be computed via some classifier now which embedding space the distance is calculated, and I see no classifier mentioned anywhere nor justified why it is useful/appropriate for that task.
>
>
> [`1] Kim, D., Woodbury, S. M., Ahern, W., Kalvet, I., Hanikel, N., Salike, S., ... & Baker, D. (2024). Computational Design of Metallohydrolases. bioRxiv, 2024-11

---

> > ### Author Response · Authors · 2024-11-27
> > **Added Motif Scaffolding Experiment in Figure 14**
> >
> > To address the feedback that _"...some of the experiments now acknowledged as interesting by the authors...should be added in order to make the work more convincing and stronger."_, we've **added the motif scaffolding experiment in Appendix Figure 14**. Unlike RFDiffusion, PLAID also generates residue identities and side chain positions. We think this is an exciting demonstration -- while we maintain that conditioning by function labels & organisms should not be discounted, we hope this experiment demonstrates the versatility of PLAID.
> >
> > We hope that this experiment, in addition to our previous comment (which clarifies that we have already provided the requested designability metrics in Table 2), have addressed your remaining concerns about accepting this paper. If this is the case, would you consider improving your score? Thank you for taking the time to engage with our work.

---

> ### Author Response · Authors · 2024-11-26
>
> Thanks for these comments and encouragement on the novelty of our approach.
>
> 1. I think we are in agreement that for unconditional design, the best validation is to see if it can be manufactured in the wet-lab; and in the absence of that, we need to look at _in silico_ metric that seem to correlate with it. **Distributional conformity has been shown in literature to correlate with real-world expressibility. WJS [1] achieved 70% expressibility using distributional conformity scores**, which at the time was "the highest reported binding rate of any antibody design method applied to trastuzumab CDR H3 redesign." We therefore think that this is a valid metric to at least consider, if not prioritize. Designability metrics have heavy bias towards samples in-distribution with what ProteinMPNN/etc. are trained on; biophysical parameter do not have this same bias. We hope this better describes our rationale behind this metric and can put us in better agreement. Furthermore, even if we examine only designability, PLAID achieves the best results of all other all-atom methods. If there are metrics that the Reviewer think is explicitly missing, please let us know.
>
> 2. As mentioned, having to model side chain positions adds up to $4^L$ additional degrees of freedom, which is non-trivial. If we only wanted to solve the Multiflow setting of backbone atoms + residue identity, that would give much more leeway to method design. **Re: side chain capabilities: Updated Figures 9 and 12 shows several cases of precise ion coordination, such as learning the tetrahedral geometry of cysteine residues coordinating the iron ion, learning the DHDH motif for zinc binding, and more**. All of these side chain positions were directly produced by PLAID. This capability actually directly mirrors the recent preprint referenced by the reviewer [2]. Are there any other specific experiments that the Reviewer might want to see on this front?
>
> 3. Re: ProteinMPNN fallibility: one of the major motivations for our work is to move away from reliance on these tools, since it compounds errors, increases installation dependencies, is slower, etc. Our point is not to say that ProteinMPNN is not a good tool -- it's of course been successfully and widely used -- but as a field, there's benefit to not putting all of our eggs in one basket. It keeps the field more nimble for progress. The point we wanted to convey is that many decisions made here were very intentionally, including why we do not use ProteinMPNN for inverse folding.
>
> 4. Re: Frechet and Sinkhorn Distances: the Frechet Distance and Sinkhorn Distance both provide a means of characterizing distance between two high-dimensional quantities. Typically this is used to assess distance between embeddings, for e.g. Inception embeddings, but since in our work, we are directly generating latent embeddings, we just directly calculate the distance in this space, i.e. in the CHEAP latent space. We will update this in the manuscript to make it more clear, along with the other fixes suggested.
>
> 5. On motif scaffolding: we maintain that **not centering our method on motif scaffolding is a strength rather than weakness, as it increases the surface area of what we can examine as a field**. We're happy to implement motif scaffolding in PLAID since that has been important to the Reviewer, but we want to re-highlight the importance of the conditioning tasks we choose:
>
> * Organism expressibility is very important; not being able to express your antigen or binder in a given system is an important bottleneck in scientific discovery
> * The goal of motif scaffolding is ultimately to preserve function, which GO terms also provide definition for (and describe along a different axis)
> * To transfer a function from one organism to another, motif scaffolding could fail if the motif is different in the target organism; our compositional conditioning approach tackles this.
>
> We appreciate you engaging with our paper. If this discussion has helped us reach better agreement on our contributions, it would be great if you can improve your score.
>
> [1] Protein Discovery with Discrete Walk-Jump Sampling. https://arxiv.org/abs/2306.12360
>
> [2] Computational Design of Metallohydrolases. https://www.biorxiv.org/content/10.1101/2024.11.13.623507v1

---

### Author Response · Authors · 2024-11-24
**Response to All Reviewers**

We’re grateful for the constructive feedback from Reviewers. All reviewers agree on the novelty, and significance of enabling better access to input data and harnessing information in pretrained model weights. With scaling laws trends [1], increasingly efficient attention kernels [2], and protein folding models expanding to output more modalities [3,4,5], we expect this paradigm to become even more pertinent.

We’ve added experiments and manuscript changes to address the Reviewers’ concerns. A point-by-point response to each review is below. As an overview of our revisions:

1. To better analyze capability differences between PLAID and existing baselines, **we visualize and compare performance at different lengths up to 512 (Updated Figure 5, Table 2, 3, Appendix Figure 9)**. This shows that **for longer sequences, PLAID better balances quality and diversity**. When visualized in **Updated Figure 5**, we see that existing methods bias towards alpha helices and demonstrates severe mode collapse at $L=256$. This is in addition to the better distributional conformity to natural sequences we originally observe, which is shown in [6] to be highly important for experimentally-realizable proteins (Updated Figure 4A).
2. **We’ve greatly expanded our case studies of function conditioning case studies.** In **Updated Figure 6**, we observe that generations have conserved catalytic residues involved in iron binding, kinase activity, deaminase activity, heme binding, and more, while maintaining low sequence identity (ensuring novel designs). Furthermore, transmembrane proteins consistently place hydrophobic residues at the core and hydrophilic residues at the ends, as expected. We hope this addresses conditioning utility concerns from Reviewers bWWc and WZZz; this is an exciting demonstration that PLAID all-atom generations can achieve great precision.
3. Reviewers bWWc and 4icF reference Multiflow; we’ve updated the manuscript to more clearly convey that **Multiflow only generates backbone atoms, and is not an all-atom generation method.** If we assume each side chain to have 0 to 4 rotamers, this induces $4^L$ more degrees of freedom. This also means that it cannot mirror PLAID's ability to capture fine-grained details for function, such as how side chains mediate catalysis (Figure 6). Additionally, the Multiflow paper does not address conditioning; PLAID is instead designed around the availability of annotations and controllability. It should be noted that **amongst all-atom generation methods, PLAID achieves state-of-the-art cross-modality consistency**. For completeness, wherever possible, we retained comparisons to Multiflow.
4. Some reviewers (bWWc, WZZz) have expressed concern with our choice of conditioning by function and organism rather than motif scaffolding. This is an exciting direction for future work, and the PLAID paradigm is fully compatible with motif scaffolding, by fixing the input sequence at inference time. We will include this experiment for the camera-ready version. Furthermore, we hope **Updated Figure 9** case studies will be of interest.
5. We wish to highlight that the primary motivation for our work is not just the model and its performance, but to describe a **paradigm for multimodal generation by learning the latent space of a predictor from a more abundant data modality to a less abundant one**. This is also why we use GO terms, as a proxy for the vast quantities of natural language annotations available in sequence databases. Whenever possible, we chose architectures and techniques that can be easily generalized to new models beyond ESMFold.

We’ve made **major enhancements to the main submission PDF**, including:
* Architectural schematic of DiT and conditioning **(Updated Figure 3)**
* additional experiments and explanations of ablations and sampling speed **(Updated Table 1, 4)**
* significantly expanded number of case studies on active site conservation of generations
* t-SNE sanity check of organism conditioning **(Updated Figure 10B)**
* using Sinkhorn distance rather than Frechet Distance to assess distances between generated and real latent distributions **(Updated Figure 10D)**
* rewriting the Methods for coherency

We hope reviewers can take a new look at our updated manuscript, and we warmly welcome additional comments.

[1] Scaling Laws for Neural Language Models. (Kaplan et al., 2020)

[2] FlashAttention-3: Fast and Accurate Attention with Asynchrony and Low-precision. (Shah et al., 2024)

[3] Generalized biomolecular modeling and design with RoseTTAFold All-Atom. (Krishna et al., 2024)

[4] Accurate structure prediction of biomolecular interactions with AlphaFold 3. (Abramson et al., 2024)

[5] Boltz-1: Democratizing Biomolecular Interaction Modeling. (Wohlwend et al., 2024)

[6] Protein Discovery with Discrete Walk-Jump Sampling. (Frey et al., 2023)

[7] Simulating 500 million years of evolution with a language model. (Hayes et al., 2024)

---

### Meta-Review · Area_Chair_25v8 · 2024-12-19

**Metareview:**

The authors propose PLAID (Protein Latent Induced Diffusion), which generates multimodal biological data by learning from abundant sequence data to predict less abundant structural data. Focused on all-atom structure generation, PLAID produces both 3D structures and 1D sequences, with emphasis on sidechain atom placement. By using sequence-only training, PLAID expands the sampleable data distribution and enables conditioning on additional annotations, such as Gene Ontology keywords and organism data. Despite lacking structure inputs during training, PLAID achieves strong performance in structure quality, diversity, novelty, and cross-modal consistency.

### Strengths:

1. Leveraging the latent space of existing structure prediction models for protein structure (all-atom) and sequence co-design is a novel and innovative approach.

2. The use of only sequence data for training the diffusion model enables scalability to larger datasets and the potential to leverage more diverse training data.

### Weaknesses:

1. The model underperforms significantly in terms of cross-modal consistency and diversity compared to Multiflow. Although the authors claim that Multiflow can only generate backbone atoms, transforming it into an all-atom version is straightforward, for example, by using a two-stage training process like Chroma [1] or by adding a side-chain packing step. The weak performance relative to Multiflow reduces the impact of this paper. The authors should at least compare their approach to a trivial all-atom version of Multiflow.

2. There are some inaccuracies in the paper. For instance, Chroma [1] can generate all-atom structures, even though it is not end-to-end. However, the authors incorrectly classify it as a non-all-atom generation method.

### Overall:

This is a borderline paper. While the idea is novel and promising, the current version is not yet suitable for publication due to the relatively weak results and lack of sufficient comparisons.

[1] Illuminating protein space with a programmable generative model, Nature 2023.

**Additional Comments On Reviewer Discussion:**

During the rebuttal period, the authors provided additional experimental results and explanations, including an analysis of the capability differences between PLAID and existing baselines, expanded case studies, comparisons with Multiflow, and justification for the choice of conditioning by function and organism. Most concerns have been addressed. However, I believe the explanations for why the proposed method underperforms Multiflow are still insufficient. The authors should implement an all-atom version of Multiflow to better support the claimed benefits of all-atom generation.

---

### Decision · Program_Chairs · 2025-01-22

Reject